# Rapid room-temperature phosphorescence chiral recognition of natural amino acids

Xiaoyu Chen[1], Renlong Zhu[1], Baicheng Zhang [1], Xiaolong Zhang[1], Aoyuan Cheng[1], Hongping Liu[1], Ruiying Gao[2], Xuepeng Zhang [1], Biao Chen [1] ✉, Shuji Ye [1], Jun Jiang [1] & Guoqing Zhang [1,3] ✉

Chiral recognition of amino acids is very important in both chemical and life sciences. Although chiral recognition with luminescence has many advantages such as being inexpensive, it is usually slow and lacks generality as the recognition module relies on structural complementarity. Here, we show that one single molecular-solid sensor, L-phenylalanine derived benzamide, can manifest the structural difference between the natural, left-handed amino acid and its right-handed counterpart via the difference of room-temperature phosphorescence (RTP) irrespective of the specific chemical structure. To realize rapid and reliable sensing, the doped samples are obtained as nano-crystals from evaporation of the tetrahydrofuran solutions, which allows for efficient triplet-triplet energy transfer to the chiral analytes generated in situ from chiral amino acids. The results show that L-analytes induce strong RTP, whereas the unnatural D-analytes produce barely any afterglow. The method expands the scope of luminescence chiral sensing by lessening the requirement for specific molecular structures.

Room-temperature phosphorescence (RTP) of organic phosphors, particularly from guest-host doped systems, has become a burgeoning research area in recent years, with the benefits of long-lived lifetimes and modulation flexibility from structurally versatile chemical entities such as organic molecules and polymers[1–4]. Guest-host RTP systems have made significant advancements in applications of various fields, including next-generation optoelectronics, high-contrast bioimaging, chiral recognition, anti-counterfeiting and optical sensors[5–15]. Recently, there is growing attention paid to the design and structure-property relationship investigation on RTP systems with chiral moieties, e.g., circularly polarized luminescence (CPL)[16–25]. Since chirality is an integral part of nature, more spectroscopic methods for understanding how molecular chirality, excited state, and electron spin are correlated will help elucidate fundamental physical principles and bring about innovative technological changes[26–28].

More recently, we observed chiral-selective phosphorescence enhancement (CPE) in a donor-acceptor system, which largely stems from chirality-dependent energy transfer (CDET): more efficient energy transfer occurs from the triplet excited-state phthalimide (PI) to the ground-state naphthalimide (NI) if the chirality of the energy donor and that of the acceptor are the same[29]. If the CDET process were universal, i.e., the probability of energy-transfer being more sensitive to chirality vs. structural complementarity, then one could a priori use one single energy donor to differentiate the chirality of many different chiral structures attached to the same energy acceptor, which manifests the energy-transfer efficiency by emitting stronger or weaker phosphorescence "afterglows", respectively. However, the reported PI/NI system suffers a major drawback as the reaction between the chiral amine and the π-conjugated phosphor is too harsh (e.g., reflux at high temperature in acidic solutions) to allow for facile construction of the chiral donor-acceptor systems. Here, we present a more universal sensing scheme to allow for rapid chiral recognition with RTP (Fig. 1a): the amino acid and the highly reactive naphthoyl chloride can under ambient conditions easily be converted into a chiral

[1]Hefei National Research Center for Physical Sciences at the Microscale, University of Science and Technology of China, Anhui 230026 Hefei, China. [2]School of Chemistry and Materials Science, University of Science and Technology of China, Hefei, Anhui 230026, China. [3]Hefei National Laboratory, University of Science and Technology of China, Hefei, Anhui 230094, China. ✉e-mail: biaochen@ustc.edu.cn; gzhang@ustc.edu.cn

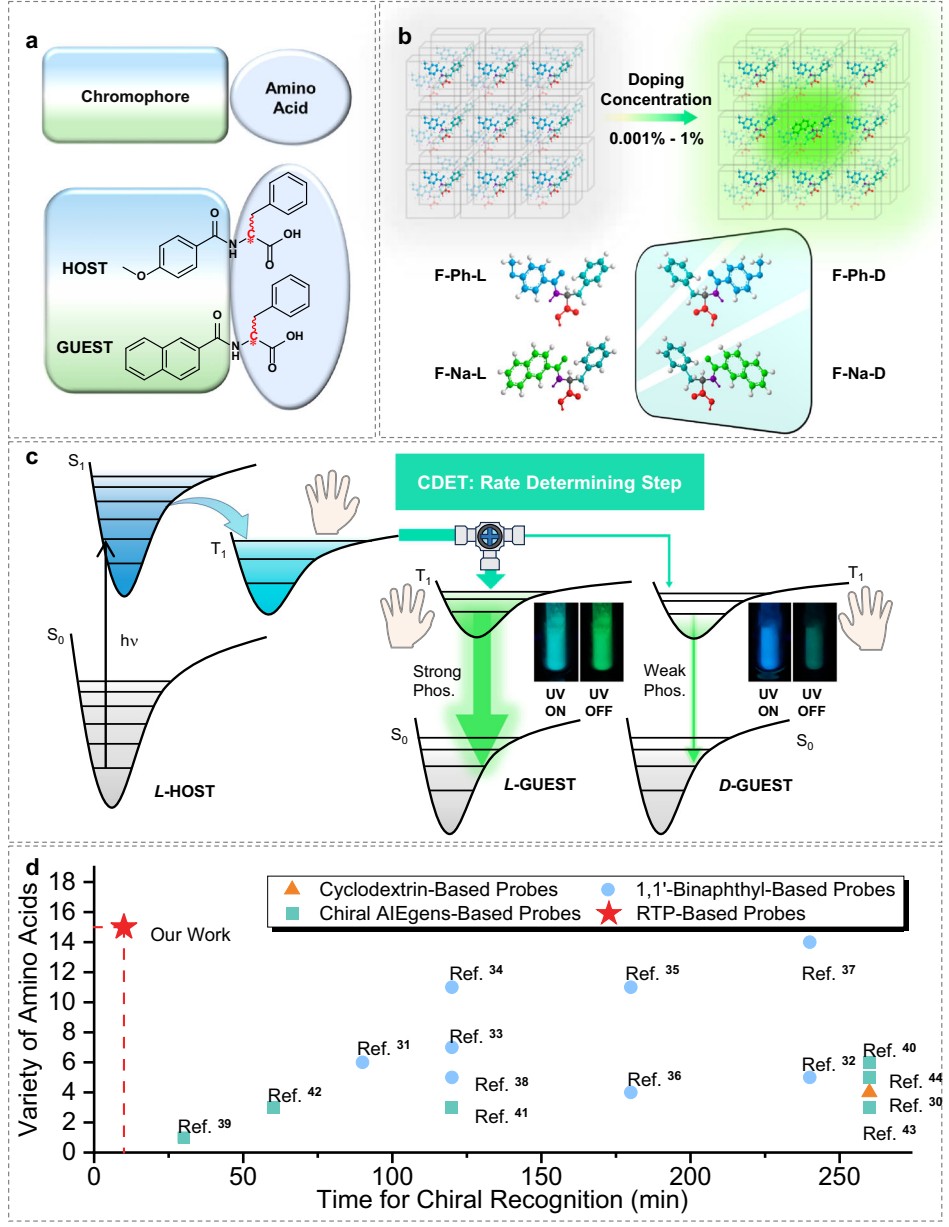

**Fig. 1 | Underlying photophysical principles for constructing chiral room-temperature phosphorescence (RTP) system utilizing amino-acid building blocks. a** Generalized bimodular design strategy constituting a chiral amino acid residue and an aromatic chromophore (top); an exemplar guest-host RTP system based on such molecular design (bottom). **b** Artist's impression of guest doping induced RTP with workable concentration ranges. **c** Proposed schematic diagrams for optimal chiral-selective room-temperature phosphorescence enhancement (CPE) from the best combination of molecular building blocks. ISC: intersystem crossing; CDET: chirality-dependent energy transfer; higher excited states $S_n/T_n$ and internal conversion (IC) are omitted for clarity. Inset: photographs of **F-Na-L** (left) and **F-Na-D** (right) in **F-Ph-L** during and immediately after 254 nm light irradiation at 77 K, respectively. **d** A survey of luminescence chiral sensing for amino acids (chiral recognition ratios ≥ 3.0[55]) indicates that the current strategy has the best performance in both sensing time and substrate variety among all published studies. (Sensing systems that do not list response time[30,40,43,44] are listed at t = 260 mins).

energy accepting phosphor, which is used to sensitize RTP from the L-phenylalanine-based triplet energy donor medium (Fig. 1b). The L-phenylalanine derivative was selected as the universal triplet energy donor because it can be produced in large batches with relatively easy purifying method (crystallization). Firstly, the CDET process (Fig. 1c) was characterized by synthesizing L-phenylalanine-based energy donor (**F-Ph-L**, Fig. 1b) and acceptor (**F-Na-L**) molecules. Then, a CPE chiral sensing protocol was developed using the **F-Ph-L** molecular-solid material, fabricated as nanocrystals to reduce sensing time and to maximize signal intensity, as the solo energy donor medium and tested for the CPE ratio between **F-Na-L** and its enantiomer **F-Na-D** as donors.

Finally, all 15 chiral natural amino acids and their unnatural enantiomers were screened with the established protocol, making the method the best among all published luminescence chiral sensing studies with sensing times as short as a few minutes (Fig. 1d)[30–44].

## Results

### Design and discovery

As an initial proof of concept, four chiral amino acid derivatives were first synthesized between two acyl chlorides and chiral phenylalanine (F) derivatives (Supplementary Fig. 1) and rigorously purified. Specifically, the two chiral amino acids (D and L) were chemically modified

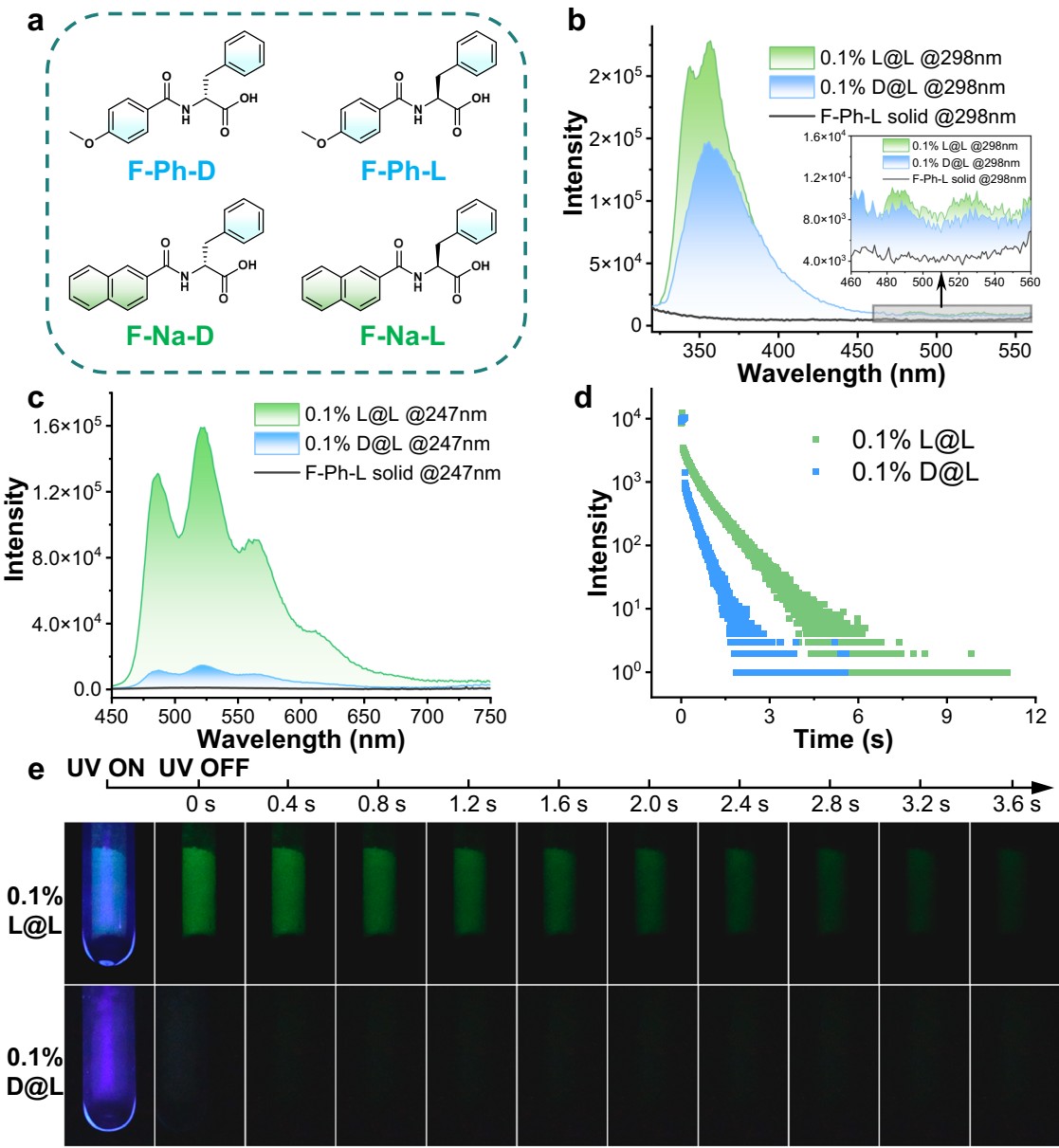

**Fig. 2 | Chiral-selective room-temperature phosphorescence enhancement (CPE) for phenylalanine derivatives. a** Chemical structure of four chiral phenylalanine-modified compounds for constructing doped RTP systems. **b** Steady-state photoluminescence (PL) spectra of the host matrix and the doped samples at 298 K ($\lambda_{ex}$ = 298 nm). **c** Delayed emission (DE, $\Delta t$ = 5 ms) spectra of the host matrix and the doped samples at 298 K ($\lambda_{ex}$ = 247 nm). **d** Time-resolved RTP emission for the two doped samples. ($\lambda_{ex}$ = 280 nm from the spectraLED-280 and $\lambda_{em}$ = 526 nm) **e** Photographs of combinations of the doped samples during and immediately after 254-nm light irradiation at 298 K. (The guest-to-host ratio is 0.1% for all samples in the solid state).

into **F-Na-D** and **F-Na-L** (Fig. 2a) by using naphthamide as the excited-state energy accepting moiety (guest) and benzamide as the energy donor (**F-Ph-D** and **F-Ph-L**, host), the four of which were characterized by [1]H and [13]C nuclear magnetic resonance (NMR) spectra, high resolution mass spectrometry (HRMS) (Supplementary Figs. 60–74) and elemental analysis (EA). The model compounds were as optical isomer pairs with high purity (>99.5%) and high enantiomeric excess (ee) values (>99.2%), which were verified by chiral high-performance liquid chromatography (CHPLC, Supplementary Figs. 3 and 4). The circular dichroism (CD) spectra (Supplementary Fig. 9) also support their chiral purity with an absolute CD signal of 5-15 mdeg (0.04 mM for **F-Ph** and 0.08 mM for **F-Na**). The absorption (Supplementary Figs. 7 and 8) and emission spectra (Supplementary Fig. 10a and b) show that the obtained amino acid derivatives display no RTP in dilute solutions (2-methyltetrahydrofuran, 2-Me-THF) or in the solid state individually,

while long-lived phosphorescence only emerges at the liquid-nitrogen temperature (77 K, Supplementary Figs. 10c–f). The radiative decay from the triplet excited state ($T_1$) to the ground state ($S_0$) is extremely inefficient presumably due to small spin-orbit coupling (SOC) for the $T_1 \rightarrow S_0$ transition, which is largely [3]π-π* in nature. Consequently, phosphorescence lifetimes on the order of a few seconds can be observed when nonradiative decay processes are suppressed[45]. However, long-lived RTP is produced when the benzamide molecular-solid host (**F-Ph**) is doped with the naphthamide guest (**F-Na**) at a concentration as low as 10 parts per million (ppm), although the mechanism for doping-induced RTP remains elusive[46–49].

**Chiral-selective RTP enhancement for phenylalanine derivatives**
The doped chiral RTP samples were readily fabricated via direct solvent evaporation without additional engineering processing, since solution-

processed molecular assemblies retain their chirality in the solid-state according to a method by Beard and Luther et al.[50] (see SI for a detailed description of sample preparation for rigorous spectroscopic investigations). The chiral amino acid derivative **F-Ph-L** was initially examined as the host material, which is not photoluminescent in the solid state at room temperature. The pure guest **F-Na-L** solid, on the other hand, is strongly fluorescent but not phosphorescent under the same conditions (Supplementary Figs. 10 and 11 and Supplementary Table 1). However, deep blue fluorescence (*circa* 357 nm, τ = 7.65 ns) and green RTP (*circa* 526 nm, τ = 637 ms) arise simultaneously leading to visibly white photoluminescence when the **F-Na-L** (0.1%, w/w) guest is mixed with the **F-Ph-L** (**L@L**, Fig. 2b) host. Surprisingly, when the other enantiomer (**F-Na-D**) is applied to the **F-Ph-L** host medium, weaker fluorescence intensity occurs with even weaker RTP intensity ratio. Delayed photoluminescence emissions (DE, Δt = 5.0 ms) show that the enantiomeric RTP enhancement ratio (which is defined as $ep_{RTP} = I_{L@L}/I_{D@L}$) is 9.3 (Fig. 2c), and the **L@L** sample also exhibits a visually distinguishable longer time (Fig. 2d, Table 1). It has to be note that, compared with the pure guest-solid, the doped samples show blue-shifted guest emission spectra, which is ascribed to a lack of guest dimeric interactions (e.g., exciton splitting[51]) capable of lowering the exciton energy.

Therefore, we have shown that the enantioselective discrimination for amino acid-based system could be quantified via RTP spectroscopy under rigorous test conditions. We next show that the host-to-guest energy transfer process is key to realizing enantioselective discrimination. The process was clearly validated by the quenching of host emission and a substantial shortening of host lifetime (Supplementary Fig. 13 and Supplementary Table 2), where solid sample mixtures of the same chirality (e.g., **L@L**) quench host photoluminescence more effectively than those consisting of molecules with opposite chirality (e.g., **D@L**), confirming the CDET process shown in Fig. 1c. In addition, we compared the 2D excitation-emission-intensity spectra of **L@L** and **D@L** solids (the solid-state absorption spectra are provided in Supplementary Fig. 8), respectively (Supplementary Fig. 14), where it is evident that CDET ceases to exist for the **D@L** sample (Supplementary Fig. 14b showing that RTP intensity diminishes at higher photon energy mainly absorbed by the host solid). The 2D excitation-emission-ep map also supports the conclusion (Supplementary Fig. 14c): ep of RTP is significantly enhanced when the host solid is photo-excited (240-300 nm) rather than direct excitation (300-350 nm) of the guest molecule. When the wavelength-dependent ep of RTP is plotted against excitation intensity (Supplementary Fig. 14d), the function coincides with the spectrum of the solid-state absorption of the L host (Supplementary Fig. 15). To verify the symmetrical condition (i.e., **D@D** and **L@D**) for the experiment, **F-Ph-D**, the enantiomeric host of **F-Ph-L**, is also found to achieve enantioselectivity as shown in Supplementary Fig. 16, where a mirror-image-like relationship of RTP responses is observed toward the enantiomers of the **F-Na** guests. From the RTP spectra, the $ep_{RTP}$ ($I_{D@D}/I_{L@D}$) value of 9.5 with a lifetime of 649 ms for **D@D** (Table 1) confirms the consistency of the chiral discrimination method using the amino acids-based guest-host system under the rigorous experimental condition. Conversely, enantiomeric guest dopants in a racemic host display almost no spectral and lifetime discrimination (Table 1, Supplementary Fig. 16), attesting to the mechanistic validity of the CPE process.

### Influences of different guest-to-host ratios in solid-state samples

We next investigated how the guest-to-host content ratio influences enantioselectivity via both fluorescence and RTP and how medium chirality influences fluorescence and RTP ep values, using the same **F-Ph** and **F-Na** exemplar pair under rigorous experimental conditions. The PL spectra are presented in Fig. 3a and Supplementary Figs. 19 and 20: when the host and guest molecules have the same chirality (i.e., **L@L** or **D@D**), stronger photoluminescence emerges for both the fluorescence band (monitored at the wavelength range of 330-420 nm) and the RTP

band (range of 450-700 nm) compared to solid mixtures consisting of molecules with opposite chiralities (**D@L** or **L@D**). For fluorescence, the multiples of enhancement are limited, ranging from ep = 1.6 to 3.2 for samples with reliable signal-to-noise ratios (≥0.01%). However, the RTP spectra (Fig. 3b and Supplementary Figs. 19 and 20) show higher ep values under the same conditions (≥6.0, Fig. 3c). The disparity in ep values between fluorescence and RTP is attributed to the fact that guest fluorescence can occur via both Förster and Dexter types of energy transfer while guest RTP is only limited to the latter type. We then obtained wavelength-dependent ep values for fluorescence emission as well, and found that these values are independent of excitation energy (Supplementary Fig. 21), clearly indicating that the long-ranged dipole-dipole Förster process is not sensitive to chirality. Based on these experiments results, we can then deduce that the shorter-ranged Dexter energy transfer, which is the sole energy transfer mode for RTP, is responsible for such enhanced ep differences. Time-resolved emission spectra also show that the samples with the same chirality have longer lifetimes compared to the opponents (Supplementary Figs. 22–24 and Supplementary Tables 3–5). In contrast, enantiomeric guest molecules embedded in the racemic host (**F-Ph-DL**) medium display less pronounced spectral and lifetime discrimination with ep values ≤ 2.0 (Fig. 3d). It is noted that the ep value of a guest-host doping concentration of 10 ppm is ≤2.0, since the RTP signal-to-noise ratio is too low to be reliable. Additionally, when tested at 77 K with suppressed nonradiative transitions, the same spectral reliability is restored with prominent chiral discrimination again observed (Supplementary Figs. 25–27), where better visual differentiation is shown in Supplementary Fig. 28 and the CIE Figure in Supplementary Fig. 29.

### Exploration of morphology and microstructure

Since the solid-state luminescence is highly sensitive to medium morphology, we also investigate how different processing methods might influence the surface microstructures. As can be seen from the scanning electron microscope (SEM) images (Fig. 4a–d) and the power X-ray diffraction (PXRD) patterns (Supplementary Fig. 30), doping with guest molecules of different chirality have no observable effect on the microscopic morphologies for solid-state samples obtained from either tetrahydrofuran (THF, Fig. 4a, b) or a mixture of chloroform/n-hexane (CHCl$_3$/HEX, Fig. 4c, d). However, the influence of solvent is tremendous: evaporation from the THF solution produces crystals of much smaller sizes with lower aspect ratios, which in contrast to large belt-like crystals acquired from CHCl$_3$/HEX. The spectroscopic differences were also measured (Fig. 4e, f), where much stronger RTP intensity and a higher ep value are achieved for samples obtained from THF. To investigate the effect of doping on the microstructure of the host molecules, we measured the sum frequency generation (SFG) spectra ranging from 2990 cm$^{-1}$ to 3110 cm$^{-1}$ (Fig. 4g, h). The SFG spectra show a strong resonant peak at 3064 ± 3 cm$^{-1}$ and a weak shoulder peak at 3053 ± 2 cm$^{-1}$, which are assigned to the $v_2$ and $v_{7b}$ vibrational modes of the phenyl group respectively[52–54]. Both ppp and ssp SFG spectra show little changes when the guest molecules are doped, indicating that the microstructure of the host molecules remains largely the same. The study suggests that doping of guest molecules of different chiralities does not induce morphological changes in the host material.

### Development of rapid test protocol for amino acids

At this point, we have theoretically shown that CDET-based RTP disparity could be used for amino acids chiral sensing. To apply this principle in real applications, we now devise a rapid test protocol under less rigorous conditions for chiral recognition of the free amino acids L- and D-phenylalanine (Supplementary Fig. 31). The protocol only requires prefabricated molecular-solid materials from one of the two host enantiomers **F-Ph-L** or **F-Ph-D** (with the L enantiomer used as an example). As illustrated in the Figure, an aqueous NaOH solution (1 mol L$^{-1}$) and THF stock solutions of naphthoyl chloride (5 mg mL$^{-1}$)

were prepared. L- and D-phenylalanine was added into an Eppendorf tube with 0.5 ml THF. Aliquots of the two colorless solutions (NaOH and NaACl) were pipetted into the tube to instantly generate a yellow-colored solution. The yellow color quickly dissipates within 5 minutes, indicating the completion of the reaction. 100 μL of the reaction mixture was diluted with 900 μL dichloromethane and acidized with 25 μL 2 M HCl. Then, two drops (circa 50 μL) of the mixture was drop-cast onto the prefabricated glass slide loaded with 20 mg of the L-isomer of the host solid (powder or film). The sample was visibly air dried after 1 min; the glass slide was then taken for visual inspection under a 254-nm UV lamp. If a green afterglow lasting for >1 s was present, it can be determined that the unknown aliquot of the mixture solution contained L-phenylalanine; otherwise, when the afterglow was weak and RTP emission very short (<0.2 s, more on the cyan-blue in color), the isomer was then determined to be D-phenylalanine.

**Wide substrate scope for chiral recognition of amino acids (AAs)**
To show that the **F-Ph** molecular solid-material is universal for RTP chiral sensing, we applied the established protocol to other chiral amino acids shown in Fig. 5. All 19 chiral natural amino acids undergo a one-step reaction with 2-naphthoyl chloride to form guest molecules (analytes), and the ep$_{RTP}$ values for a total of 19 chiral amino acid pairs were listed, where 15 of them could be reliably discriminated with an ep value ≥ 3.0. (Detailed photographs and spectra can be found in Supplementary Figs. 32–50) It was found that amino acids with aromatic groups exhibit the best distinguishability both visually and spectroscopically (e.g., the indole group). On the other extreme, the chirality of two basic amino acids could not be discerned at all, unless a different reaction protocol was applied. The [1]H-NMR spectra reveal that in the 5 mins reaction window, no naphthoyl moiety was chemically attached to the amine group of histamine (H) and arginine (R) while the reaction was largely completely for the aromatic amino acids (Supplementary Figs. 51–57). As has been presented in Fig. 1, the chiral recognition of amino acids using photoluminescence usually requires lengthy reaction time and relatively high amino acid concentrations (e.g., 150 eq. to that of the luminescence chiral probe[37]), which poses a significant challenge when the amount of amino acid to be sensed is not abundant. The current protocol, however, requires only a tiny amount of the amino acid analyte within 5 mins, which easily outperforms all other reported protocols.

**Table 1 | Photoluminescence properties of dopant (w/w = 0.1%) samples at room temperature**

| Samples[a] | λ$_F$ [nm][b] | τ$_F$ [ns][c] | λ$_{RTP}$ [nm][d] | τ$_{RTP}$ [ms][e] | ep$_{RTP}$[f] |
|---|---|---|---|---|---|
| L@L | 357 | 7.65 | 526 | 637 | |
| D@L | 357 | 5.09 | 526 | 322 | 9.3[g] |
| D@D | 357 | 6.94 | 526 | 649 | |
| L@D | 357 | 4.25 | 526 | 454 | 9.5[g] |
| L@DL | 357 | 4.42 | 526 | 986 | |
| D@DL | 357 | 3.72 | 526 | 968 | 1.3 |

[a]Guest-host molecular solids (w/w = 0.1%);
[b]Fluorescence emission maxima of steady-state photoluminescence spectra excited at 298 nm;
[c]Apparent fluorescence lifetime (weighted average sum, nanoLED-280);
[d]RTP maxima of delayed emission spectra excited at 247 nm;
[e]Apparent RTP lifetime (spectraLED-280);
[f]enantiomeric RTP enhancement ratios (ep$_{RTP}$, I$_{L@L}$/I$_{D@L}$, I$_{D@D}$/I$_{L@D}$ or I$_{L@DL}$/I$_{D@DL}$).
[g]the average result of 3 independent trial processes. (Supplementary Fig. 12)

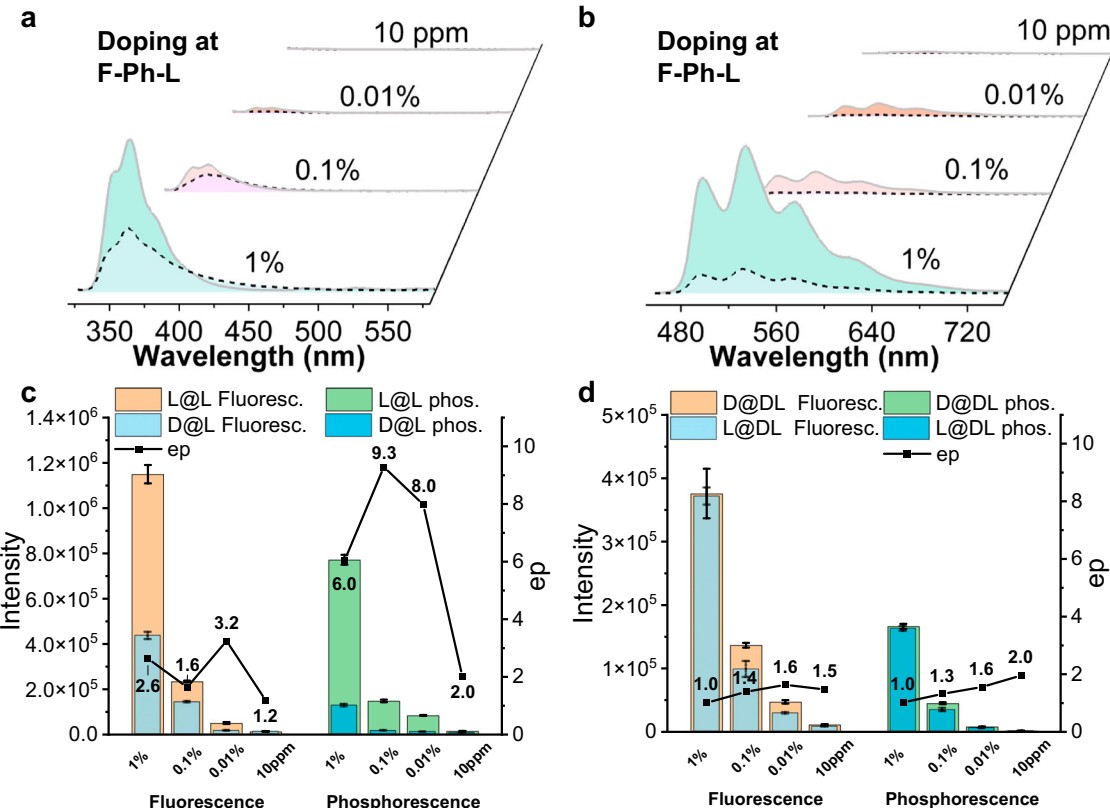

**Fig. 3 | Influences of different guest-to-host ratios in solid-state samples.**
**a** Steady-state photoluminescence (PL, λ$_{ex}$ = 298 nm) and **b** Delayed emission (DE, Δt = 5 ms, λ$_{ex}$ = 247 nm) spectra of **F-Na-L** (solid borderline) and **F-Na-D** (dash borderline) guests doped in **F-Ph-L** solid matrix (w/w = 10 ppm-1%) in air at 298 K. The photoluminescence intensity and ep values (intensity ratios of steady-state photoluminescence at 357 nm and delayed emission at 526 nm) vs. dope ratios of the guest in **c F-Ph-L** and **d F-Ph-DL** (w/w = 10 ppm −1%, values are means ± s.e.m., $n$ = 3 or 4).

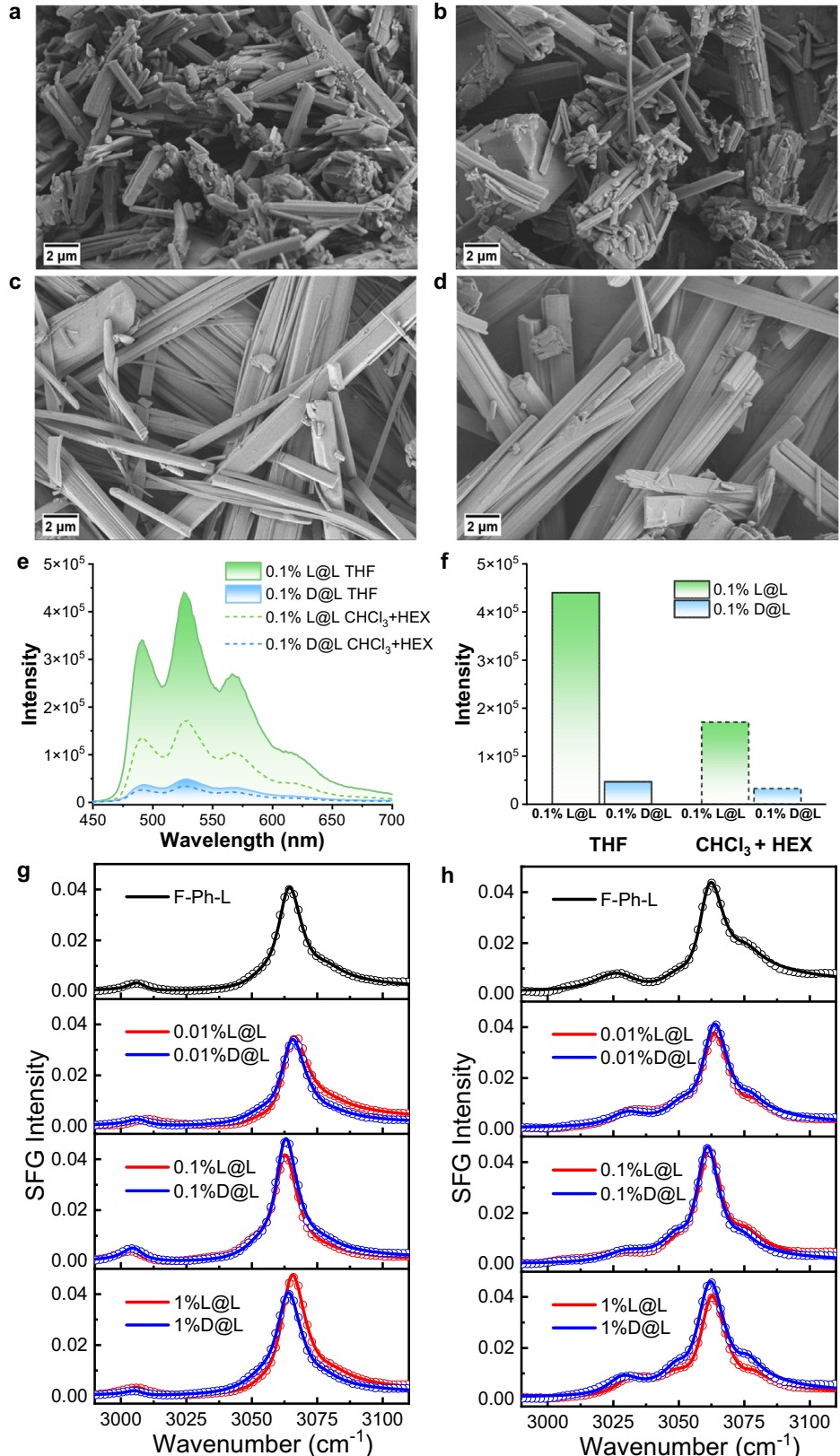

**Fig. 4 | Exploration of morphology and microstructure.** SEM images of doped guest-host samples of **a** **L@L** and **b** **D@L** obtained by recrystallization with tetra-hydrofuran (THF); **c** **L@L** and **d** **D@L** obtained by recrystallization with chloroform (CHCl₃) and n-hexane (HEX) (90:10). **e** Delayed emission (Δt = 5 ms) spectra of **L@L** and **D@L** by recrystallization with THF (solid borderline) and CHCl₃: HEX = 90:10 (dash borderline) in air at 298 K ($\lambda_{ex}$ = 247 nm). **f** The phosphorescence intensity of different doped samples for different solvent systems. (The guest-to-host ratio is 0.1% for all samples above in the solid state) **g**, **h** The sum frequency generation (SFG) spectra of the host and doped samples (w/w = 0.01–1%) with different polarization combinations: **g** ppp-SFG spectra; **h** ssp-SFG spectra.

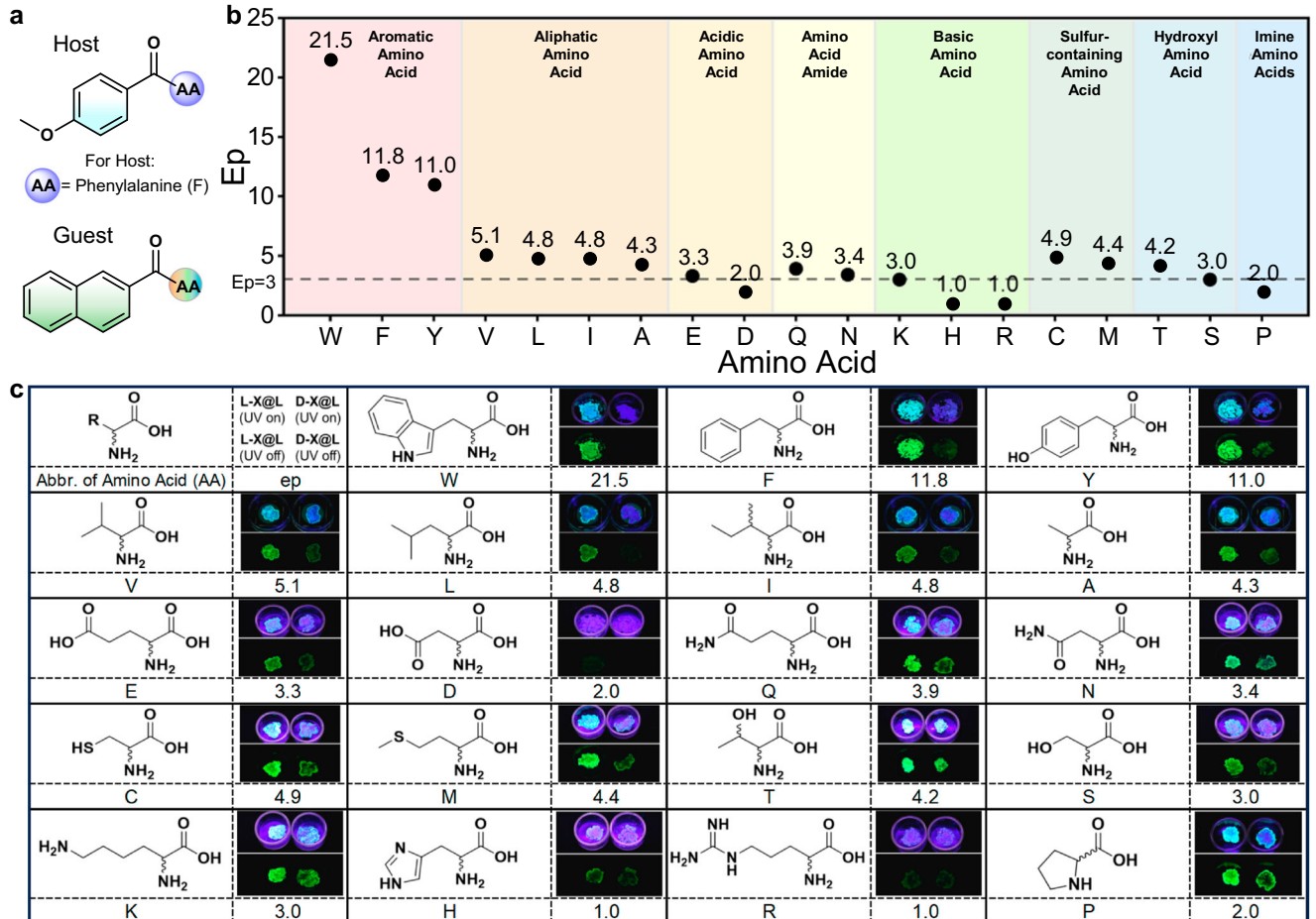

**Fig. 5 | A universal sensing material for chiral standard amino acids. a** Chemical structure of hosts and guests in chiral recognition of amino acids. **b** $Ep_{RTP}$ values for chiral amino acids grouped into different structural categories, where it is found that a total of 15 pair of chiral amino acids could be reliably distinguished (ep ≥ 3.0) while the other four failed the test protocol due to poor reactivity between the acyl chloride and the chiral amino acid based on NMR evidence. **c** chemical structures with one-letter abbreviations of the tested chiral amino acids (AAs for the guest), and photographs showing the visual RTP "afterglow" difference using the same sensing medium material under the protocol.

## Regulation of enantioselective differentiation and mechanism investigation

Finally, taking advantage of the CDET mechanism, we show that the enantioselective differentiation of the amino acids-base system can be optimized with great tunability by designing a heavy-atom-substituted guest **F-Na-Br** (Fig. 6a). As shown in Fig. 6b, the $ep_{RTP}$ ratio is also related to the phosphorescence radiative transition rate from $T_1$ to $S_0$ ($k_P$) of the guest, in addition to the rate of CDET from the host to the guest. Presumably, by introducing a heavy atom (e.g., bromine) to the guest, the rate of $k_P$ would then be increased, which could lead to higher $ep_{RTP}$ values. The two brominated chiral guests were synthesized and carefully characterized with high purity and e.e. values (Supplementary Figs. 5 and 6, and Supplementary Figs. 73–84), and were then applied to test the hypothesis. To exclude the influence of differing nonradiative decay rates ($k_{nr}$) and to obtain the intrinsic chiro-optical relationship, phosphorescence spectra were collected at 77 K. As shown in Fig. 6d, the brominated guest doped system exhibits CPE, where host-guest systems with the same chirality (i.e., **L-Br@L** or **D-Br@D**) exhibit much stronger phosphorescence than their counterparts with the opposite chirality (**D-Br@L** or **L-Br@D**). More importantly, the $ep_{RTP}$ values are 2× higher than systems doped with the unbrominated guest (Fig. 6c), and almost 4× higher than doped systems with the brominated host (Fig. 6e, f). The low $ep_{RTP}$ value of the brominated host system is probably due to excessive external heavy atoms in the host (99%),

which is attributed to the simultaneous increase in the RTP intensities of both the D- and L-guests. Not surprisingly, the systems with Br-substituted guests also possess better visual differentiation: **L-Br@L** displays strong green emission with green afterglow, whereas **D-Br@L** has weak cyan emission with blue afterglow from the host phosphorescence, indicating inefficient triplet-triplet energy transfer (Fig. 6g). The results point to the possibility of using CDET and CPE to optimize fine-tuning to achieve the best recognition conditions, which can potentially be expanded into other systems, showcasing the advantages of organic RTP sensing.

## Discussion

In summary, we have provided a universal design strategy to construct an amino acids-based chiral guest-host RTP system. The system possesses enantioselective discrimination photoluminescence performance in the solid state, especially chiral-selective phosphorescence enhancement (CPE) with a 25.8-fold intensity and a lifetime > 600 ms. By using this concept, the most rapid as well as the most diverse chiral recognition protocol of chiral amino acids is established by using one single nanocrystalline material enabled by efficient chirality-dependent energy transfer in triplet excited states. It has to be noted that significantly reduced CPE ratios (e.g., 21.5 (tryptophan) to 4.3 (alanine), 9.3 (**F-Ph-L**) to 4.0 (**F-Ph-L-2**), Fig. 5 and Supplementary Fig. 58) could be noted if the energy-transfer unit (e.g., the benzamide or the aromatic ring on the amino acid) is

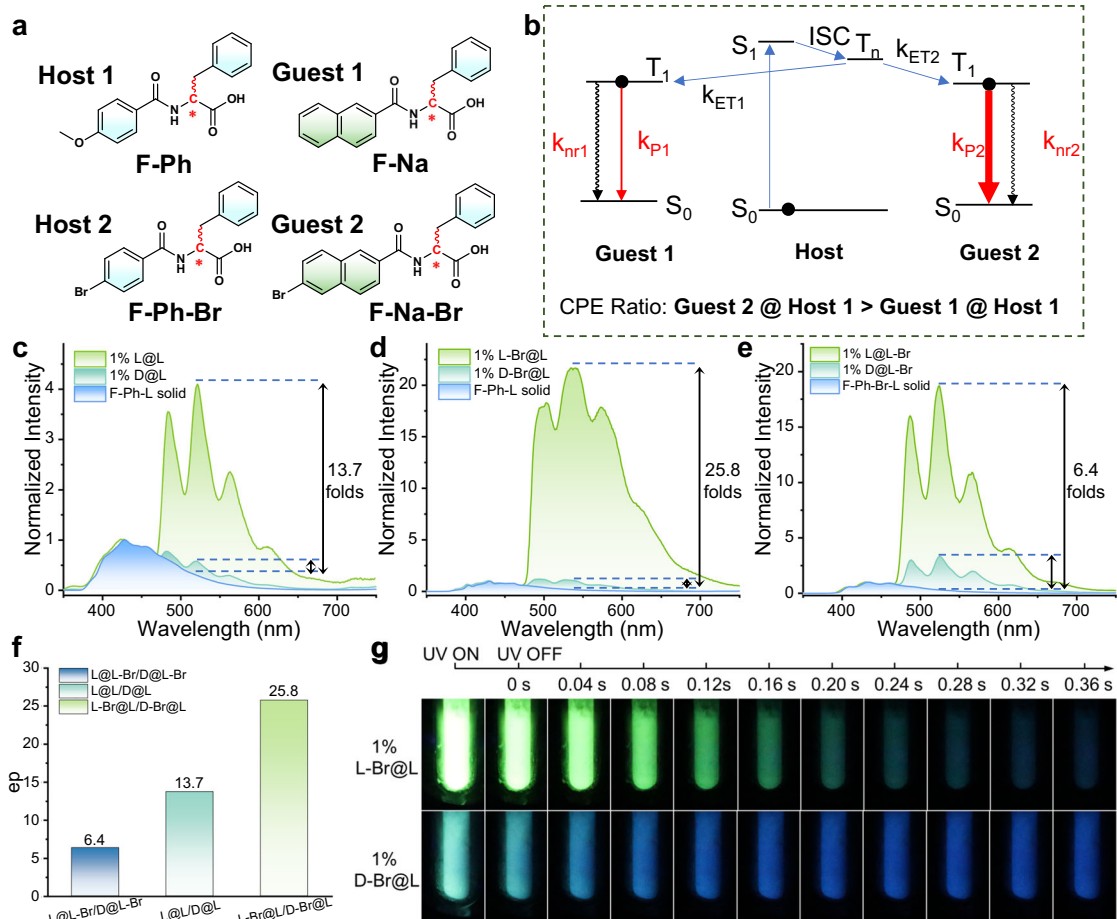

**Fig. 6 | The regulation of enantioselective differentiation via the radiative transition rate of guest molecule and the rate of energy transfer. a** Chemical structure of hosts and guests. **b** Proposed mechanism for explaining the difference in enantioselective discrimination of Guest 1 and Guest 2 (ISC: intersystem crossing; ET: energy transfer; P: phosphorescence). **c** Delayed emission (DE, Δt = 1 ms) spectra of **Guest 1** in the **L-Host 1** solid at 77 K (λ_ex = 247 nm). **d** Delayed emission (DE, Δt = 1 ms) spectra of **Guest 2** in the **L-Host 1** solid at 77 K (λ_ex = 247 nm).

**e** Delayed emission (DE, Δt = 1 ms) spectra of **Guest 1** in the **L-Host 2** solid at 77 K (λ_ex = 251 nm). **f** The ep_RTP value (intensity of delayed emission at the emission maxima) in the host matrix. **g** Photographs of combinations of **L-** and **D-Guest 2** in **L-Host 1** during and immediately after 254-nm light irradiation at 77 K, where the blue afterglow suggests less efficient energy transfer in the **D-Br@L** solid. (The guest-to-host ratio is 1% for all samples in the solid state).

absent, further validating the different theoretical foundation of the current protocol. The enantioselective differentiation capacity of the system can also be regulated by the radiation transition rate of guest molecules. We anticipate that the method could be extended to the RTP chiral recognition of other naturally occurring amino compounds.

## Methods

### Materials

Tert-butyl D-phenylalaninate hydrochloride, oxalyl dichloride, D-isoleucine and D-asparagine were purchased from Shanghai Macklin Biochemical Co., Ltd. Tert-butyl L-phenylalaninate hydrochloride and N-Acetyl-L-phenylalanine were purchased from Shanghai Aladdin Bio-ChemmTechnology Co., Ltd and N-Acetyl-L-phenylalanine was purified by recrystallization by with chloroform and n-hexane. All other reagents and solvents were obtained from Energy Chemicals and were used as received. Water was deionized with a Milli-Q SP reagent water system (Millipore) to a specific resistivity of 18.2 MΩ.cm. Analytical thin layer chromatography (TLC) was performed using glass plates pre-coated with silica gel and zinc phosphate (0.25 mm). TLC plates were visualized by exposure to UV light at 254 nm. Flash column chromatography was performed using silica gel 60 (230–400 mesh) with the indicated solvents.

### Instrumentation

NMR spectra were recorded on a Bruker AV400 NMR spectrometer at room temperature, 400 MHz for [1]H and 101 MHz for [13]C. Chemical shifts of NMR spectra are reported in ppm relative to the signals corresponding to the residual protio DMSO. Elemental analysis (EA) was performed on an Elementar Vario MICRO elemental analyzer. Electrospray ionization (ESI) mass spectra were recorded on an Acquity UPLC-Xevo G2 QT mass spectrometer (Waters). Gel filtration chromatography was performed using a chiral column (CHIRALPAK® AD-H 0.46 cm I.D. x25 cm × 5 μm, DAICEL) conjugated to an Agilent 1260 Infinite HPLC system. The absorption wavelengths used were set at 250 nm and 285 nm for **F-Ph-DL, F-Ph-L** and **F-Ph-D**, 238 nm and 270 nm for **F-Ph-Br-DL, F-Ph-Br-L** and **F-Ph-Br-D**, and 280 nm and 335 nm for **F-Na-DL, F-Na-L** and **F-Na-D**, 286 nm and 315 nm for **F-Na-Br-DL, F-Na-Br-L** and **F-Na-Br-D**. Circular dichroism (CD) spectra were recorded on a JASCO J-1500 circular dichroism spectrometer. UV-Vis absorption spectra of solutions were recorded on a PERSEE TU-1901 UV-Vis spectrometer. UV-Vis absorption spectra of solids were recorded on a UV-Visible-Near infrared Spectrophotometer - Solid 3700 DUV. The steady-state emission spectra were recorded on a Horiba FluoroMax-4 spectrofluorometer (Horiba Scientific), using a vertically mounted 150-W ozone-free cw xenon arc lamp as the light source. The delayed emission spectra were also recorded on a Horiba FluoroMax-4

spectrofluorometer, using a 10-W xenon flash lamp as the excitation source. The lifetime data were acquired with a FluoroHUB TCSPC. Lifetime data were analyzed with Data Station v6.6 (Horiba Scientific). Photographs were taken by a Canon EOS 90D camera. Quantum yield of the doped samples was measured on a Quantaurus-QY Plus UV-NIR absolute PL quantum yield spectrometer C13534-12. Scanning electron microscopy (SEM) images were taken on a Zeiss Gemini 500 Schottky Field Emission Scanning Electron Microscope at 1.0 kV. Powder X-ray diffraction patterns were recorded on a Multifunctional Rotating-anode X-ray Diffractometer - Rigaku SmartLab.

## Sample Preparation

All samples were prepared by mixing solutions of host and guest molecules, followed by evaporation under ambient conditions. Except for the mixture solvent of trichloromethane and n-hexane used in the exploration of morphology and microstructure, tetrahydrofuran was chosen as the solvent for all other samples to achieve the best spectral properties. Specifically, a 1000 μl solution with a concentration of 100.00 mg mL$^{-1}$ of the host molecule was mixed with a solution with a certain concentration of the guest molecule in a 20-mL vial. The sample as a clear solution was then allowed to evaporate under ambient conditions for several days. The crystalline solid was thoroughly dried in vacuum before optical measurements. The pure host solid sample (control) was obtained by the same procedure by adding the certain solvent without guest molecules to exclude any possible influence from solvents. For the guest solutions at lower concentrations (*i.e.*, 0.100 mg mL$^{-1}$, 0.0100 mg mL$^{-1}$, and 0.00100 mg mL$^{-1}$), serial dilution was employed. For example, a stock solution (1000 μl with a concentration of 1.00 mg mL$^{-1}$) in tetrahydrofuran was pipetted into a volumetric flask (10 mL) and dilute with tetrahydrofuran to volume, which yielded a tetrahydrofuran solution with a concentration of 0.100 mg mL$^{-1}$. The exact mass of the guest molecule can be calculated by pipetting a certain volume of the guest solution when preparing the doped samples. It's important to note that all the mixed solid samples are calculated by mass fraction. Additionally, the molecular weight of the guest molecules (**F-Na** and **F-Na-Br**) is greater than that of the host molecules (**F-Ph** and **F-Ph-Br**). Consequently, the molar fraction will be numerically smaller than the mass fraction. For instance, 10$^{-3}$ by mass ratio of **F-Na** to **F-Ph** is the equivalent of 9.4 × 10$^{-4}$ by molar ratio.

## Data availability

All relevant data generated in this study are provided in the supplementary information and also are available from the authors upon request. Source data are available. Source data are provided with this paper.

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

## Acknowledgements

We thank the Innovation Program for Quantum Science and Technology (2021ZD0303301 to G.Z.), the Fundamental Research Funds for the Central Universities (WK9990000099 to G.Z.) and the National Natural Science Foundation (21975238 to G.Z., 22273097 and 22003063 to B.C., and 22025304 and 22033007 to J.J.) for financial support. We also thank Prof. Xiao Wang from Xiamen University for his assistance in photoluminescence quantum yield measurements and Prof. Yi Luo for his helpful discussion. This work was partially carried out at the Instruments Center for Physical Science, University of Science and Technology of China.

## Author contributions

X.C., G.Z. and B.C. designed the experiments. X.C. and R.G. synthesized all compounds. X.C., A.C., and H.L. contributed to optical characterizations. X.C. and A.C. contributed to electron microscope characterizations. X.C. optimized the HPLC. R.Z. contributed to SFG characterizations. X.C., B.C. and G.Z. wrote the manuscript. X.C., G.Z. B.C., B.Z., Xi.Z., Xu.Z., S.Y. and J.J. discussed the results and edited the manuscript. B.C., and G.Z. supervised the project.

## Competing interests

The authors declare no competing interests.
