## [Peer Review File · Nature Communications]

Rapid Room-Temperature Phosphorescence Chiral Recognition of Natural Amino AcidsREVIEWER COMMENTS

Reviewer #1 (Remarks to the Author):

In this work, the authors developed a new method for the chiral recognition of amino acids by employing room-temperature phosphorescence (RTP). This method demonstrates a remarkable versatility in recognizing over a dozen of different amino acids via distinguishable afterglow at ambient conditions. The mechanism of the recognition has been well studied through elaborate spectral analysis, lifetime analysis, SEM, SFG and control experiments, which clearly clarified the roles of host, guest, and heavy-atom. This work enabled the regulation of enantioselective differentiation ability via adjusting the structure of host or guest molecules and showcased that organic phosphorescence offers comparable or even better sensitivity and potential compared to fluorescence. The manuscript was well-organized and should be of broad interest to the area of solid-state luminescence. Therefore, I recommend its publication in Nature Communications after the following minor revisions.

1. In Figure 2b and 2c, why the steady-state photoluminescence (PL) spectra and delayed emission (DE) spectra have different excitation wavelength?
2. Since the process of chiral recognition and spectral analysis occurred in the solid state, the authors should provide the solid absorption spectra of host molecules, both pre and post doping host samples are required.
3. The authors should also provide the scanning electron microscope (SEM) image of doped samples.
4. How the enantiomer excess (ee) values of the chiral analytes (amino acids) affect the ratios of chiral recognition? The authors could provide at least one example.
5. In rapid test protocol section, the characterization lacks details: what is the power of UV lamp, the distance from samples, and excitation time for the delayed emission or afterglow?
6. There are few formatting errors in manuscript and supporting information. For example: the abbreviation "CDET" needs to be defined when it first appears in the main text. In "Development of a rapid test protocol for amino acids" section, "0.5ml" and "2M HCl", there should be a space in front of the unit. The same problem also appeared in the supporting information.

Reviewer #2 (Remarks to the Author):

Zhang et al. provided a universal design strategy to construct an amino acids-based chiral guest-host RTP system. The system has photoluminescence enhanced by chiral selective phosphorescence in the solid state. This exceptional property can quickly identify the absolute configuration of amino acids. I believe the protocol is novel and interesting, and the characterization is extremely adequate, but the authors should pay attention to the following issues before the publication in Nat. Commun.

1. The authors claimed the protocol is rapid compared to other methodologies, which is determined by the reaction speed. Is there any possibility to further accelerate the reaction by adding some catalyst or reagents or modifying reaction conditions?
2. Is there any explanation for the unusual PL spectra of F-Ph-DL solid, F-Ph-D solid, and F-Ph-L solid in Figures S7d and S7f?
3. Please further explain the disparity in ϵ_p values between fluorescence and RTP. I do not receive the critical point using the Dexter-type energy transfer model. Also, the disparity in ϵ_p values between L@L and D@D cases is wondering.
4. The author believes that the energy transfer between the host and the guest is carried out through the "Dexter" mechanism, why is "Dexter" not "Förster"?
5. Why is energy transfer better in the same configuration? What is the internal relationship between energy transfer and chirality?
6. Why does the 0.1% ratio give the best ϵ_p value?
7. Why does L@L observe chiral-selective phosphorescence enhancement, while D@DL does not follow the same phenomenon, as the racemic host also contains the L configuration?
8. To demonstrate that doping brings about changes in energy transfer rather than suppression of non-radiation. The author could give the k_{nr} and k_{ISC} of D@D, D@L, and D@D.
9. The manuscript needs to be carefully examined as well as ESI. There are several corrections that need to be made, such as "Deter energy transfer requires..." → "Dexter energy transfer requires...", "naonolLED-280..." → "nanolLED-280...", "whereas D-B@L has weak..." → "whereas D-Br@L has weak..."

Reviewer #3 (Remarks to the Author):

The manuscript entitled "Rapid Room-Temperature Phosphorescence Chiral Recognition of Natural Amino Acids" provided by Zhang et al. reports a series of phenylalanine-based CPL/RTP materials. The F-Ph enantiomer (host) and F-Na enantiomer (guest) are synthesized by incorporating the original L-/D-phenylalanine moiety with phenyl unit and naphthalene, respectively. The authors emphasize that the designed host molecule (F-Ph –L) can modulate the RTP emission intensity of different chiral guest molecules (F-Na –L and F-Na –D) through the chiral catalytic triplet-triplet energy transfer. Similar studies on the generation of chiral-selective RTP based on energy transfer between host and guest have been reported by the authors' previous work (Nat Commun 14, 1514 (2023)). Accordingly, this article appears to be less innovative, which relies on similar concept for applications, while the underlying "mechanism" remains somewhat elusive. Regarding the chiral-selective amino acid recognition, what is puzzling me is why the RTP photoluminescence obtained by doping different types of chiral amino acid molecules (guests) using the same host material indeed obtained the same results. In other words, why all D/L amino acids follow the same trend whereas I do not see any "common" interaction! In my opinion, this manuscript lacks comprehensive explanations of the mechanism, which gives the impression of a collection of observations and perhaps potential applications without a solid scientific foundation. Given these considerations, I am not convinced of its suitability for publication in Nature Communications.

Other major concerns:

1. Upon reviewing Figure S7, for F-Ph-L and F-H-D, I observed the presence of an emission band within the 370-550 nm range at 77 K (Fig. S7 c-d). Moreover, there is also a delayed (5 ms) emission band within the 350-500 nm range, which is blue shifted from the previous emission at 77 K (Fig. S7 e-f). Explain.
2. The title of this manuscript is "Rapid Room-Temperature Phosphorescence Chiral Recognition of Natural Amino Acids". Additionally, at the beginning of the manuscript, the authors stated, "...To realize rapid and reliable sensing, the benzamide molecule is processed into nanocrystals by lyophilization from a mixed solvent, which allows for efficient triplet-triplet energy transfer to the chiral analytes generated in situ from chiral amino acids." It appears that rapid RTP is one of the key issues addressed in this paper, and

the morphology strongly influences the efficiency of energy transfer rates. However, the characterization of the material's microstructure is rather limited. For instance, SEM cannot identify crystal structure. Other appropriate measurements should be applied to support the mechanism proposed by the authors. Furthermore, to my knowledge, lyophilization is not necessarily used to create crystals. The experimental parameters of the lyophilization process also significantly affect the material's morphology. These parameters are critical and should be described in detail.

3. As shown in Figure 3, in addition to the RTP emission, the intensity ratio between the steady-state PL and delayed emission of fluorescence is also affected by the chirality difference between the host and guest, contrary to the authors' proposal of triplet-triplet energy transfer, as shown in Figure 1c. Explain.

4. The full name of the CDET should be presented when the term is introduced in the article for the first time.

5. In Figure 2d, the excitation wavelength, monitoring wavelength, and captured delay time should be described in the caption.

6. In "Regulation of enantioselective differentiation and mechanism verification."

Paragraph, ".....whereas D-B@L has weak cyan emission with blue afterglow from the host phosphorescence....." D-B@L should be D-Br@L.

7. I cannot find a corresponding chemical structure to the sample code "L-F-H" either in the manuscript or SI.

8. According to Figure S2, the retention time of the D-form and L-form for F-Na appears at approximately 10 min and 15 min, respectively. This is completely different from what Figure S4 displays in the HPLC spectra. However, the authors assigned both of them as the same compound (F-Na). Recheck it!

9. According to this article, the authors only provide absorption spectra in acetonitrile for all title compounds and lack solid-state absorption data. Since either RTP or CPL emission is observed in the solid state, it is necessary to provide solid-state absorption spectra.

10. It seems like the authors prepared this manuscript in a rush, where the organization, statements and labels are not well prepared, which has to be trimmed completely before submission elsewhere.

REVIEWER COMMENTS

Reviewer #1 (Remarks to the Author):

In this work, the authors developed a new method for the chiral recognition of amino acids by employing room-temperature phosphorescence (RTP). This method demonstrates a remarkable versatility in recognizing over a dozen of different amino acids via distinguishable afterglow at ambient conditions. The mechanism of the recognition has been well studied through elaborate spectral analysis, lifetime analysis, SEM, SFG and control experiments, which clearly clarified the roles of host, guest, and heavy-atom. This work enabled the regulation of enantioselective differentiation ability via adjusting the structure of host or guest molecules and showcased that organic phosphorescence offers comparable or even better sensitivity and potential compared to fluorescence. The manuscript was well-organized and should be of broad interest to the area of solid-state luminescence. Therefore, I recommend its publication in Nature Communications after the following minor revisions.

1. In Figure 2b and 2c, why the steady-state photoluminescence (PL) spectra and delayed emission (DE) spectra have different excitation wavelength?
2. Since the process of chiral recognition and spectral analysis occurred in the solid state, the authors should provide the solid absorption spectra of host molecules, both pre and post doping host samples are required.
3. The authors should also provide the scanning electron microscope (SEM) image of doped samples.
4. How the enantiomer excess (ee) values of the chiral analytes (amino acids) affect the ratios of chiral recognition? The authors could provide at least one example.
5. In rapid test protocol section, the characterization lacks details: what is the power of UV lamp, the distance from samples, and excitation time for the delayed emission or afterglow?
6. There are few formatting errors in manuscript and supporting information. For example: the abbreviation "CDET" needs to be defined when it first appears in the main text. In "Development of a rapid test protocol for amino acids" section, "0.5ml" and "2M HCl", there should be a space in front of the unit. The same problem also appeared in the supporting information.

Reviewer #2 (Remarks to the Author):

Zhang et al. provided a universal design strategy to construct an amino acids-based chiral guest-host RTP system. The system has photoluminescence enhanced by chiral selective phosphorescence in the solid state. This exceptional property can quickly identify the absolute configuration of amino acids. I believe the protocol is novel and interesting, and the characterization is extremely adequate, but the authors should pay attention to the following issues before the publication in Nat. Commun.

1. The authors claimed the protocol is rapid compared to other methodologies, which is

determined by the reaction speed. Is there any possibility to further accelerate the reaction by adding some catalyst or reagents or modifying reaction conditions?

2. Is there any explanation for the unusual PL spectra of F-Ph-DL solid, F-Ph-D solid, and F-Ph-L solid in Figures S7d and S7f?

3. Please further explain the disparity in η_p values between fluorescence and RTP. I do not receive the critical point using the Dexter-type energy transfer model. Also, the disparity in η_p values between L@L and D@D cases is wondering.

4. The author believes that the energy transfer between the host and the guest is carried out through the "Dexter" mechanism, why is "Dexter" not "Förster"?

5. Why is energy transfer better in the same configuration? What is the internal relationship between energy transfer and chirality?

6. Why does the 0.1% ratio give the best η_p value?

7. Why does L@L observe chiral-selective phosphorescence enhancement, while D@DL does not follow the same phenomenon, as the racemic host also contains the L configuration?

8. To demonstrate that doping brings about changes in energy transfer rather than suppression of non-radiation. The author could give the k_{nr} and k_{ISC} of D@D, D@L, and D@D.

9. The manuscript needs to be carefully examined as well as ESI. There are several corrections that need to be made, such as "Deter energy transfer requires..." → "Dexter energy transfer requires...", "naonOLED-280..." → "nanOLED-280...", "whereas D-B@L has weak..." → "whereas D-Br@L has weak..."

Reviewer #3 (Remarks to the Author):

The manuscript entitled "Rapid Room-Temperature Phosphorescence Chiral Recognition of Natural Amino Acids" provided by Zhang et al. reports a series of phenylalanine-based CPL/RTP materials. The F-Ph enantiomer (host) and F-Na enantiomer(guest) are synthesized by incorporating the original L-/D-phenylalanine moiety with phenyl unit and naphthalene, respectively. The authors emphasize that the designed host molecule (F-Ph -L) can modulate the RTP emission intensity of different chiral guest molecules (F-Na -L and F-Na -D) through the chiral catalytic triplet-triplet energy transfer. Similar studies on the generation of chiral-selective RTP based on energy transfer between host and guest have been reported by the authors' previous work (Nat Commun 14, 1514 (2023)). **Accordingly, this article appears to be less innovative, which relies on similar concept for applications, while the underlying "mechanism" remains somewhat elusive. Regarding the chiral-selective amino acid recognition, what is puzzling me is why the RTP photoluminescence obtained by doping different types of chiral amino acid molecules (guests) using the same host material indeed obtained the same results. In other words, why all D/L amino acids follow the same trend whereas I do not see any "common" interaction!** In my opinion, this manuscript lacks comprehensive explanations of the mechanism, which gives the impression of a collection of observations and perhaps potential applications without a solid scientific foundation.

Given these considerations, I am not convinced of its suitability for publication in Nature Communications.

Other major concerns:

1. Upon reviewing Figure S7, for F-Ph-L and F-H-D, I observed the presence of an emission band within the 370-550 nm range at 77 K (Fig. S7 c-d). Moreover, there is also a delayed (5 ms) emission band within the 350-500 nm range, which is blue shifted from the previous emission at 77 K (Fig. S7 e-f). Explain.
2. The title of this manuscript is "Rapid Room-Temperature Phosphorescence Chiral Recognition of Natural Amino Acids". Additionally, at the beginning of the manuscript, the authors stated, "...To realize rapid and reliable sensing, the benzamide molecule is processed into nanocrystals by lyophilization from a mixed solvent, which allows for efficient triplet-triplet energy transfer to the chiral analytes generated in situ from chiral amino acids." It appears that rapid RTP is one of the key issues addressed in this paper, and the morphology strongly influences the efficiency of energy transfer rates. However, the characterization of the material's microstructure is rather limited. For instance, SEM cannot identify crystal structure. Other appropriate measurements should be applied to support the mechanism proposed by the authors. Furthermore, to my knowledge, lyophilization is not necessarily used to create crystals. The experimental parameters of the lyophilization process also significantly affect the material's morphology. These parameters are critical and should be described in detail.
3. As shown in Figure 3, in addition to the RTP emission, the intensity ratio between the steady-state PL and delayed emission of fluorescence is also affected by the chirality difference between the host and guest, contrary to the authors' proposal of triplet-triplet energy transfer, as shown in Figure 1c. Explain.
4. The full name of the CDET should be presented when the term is introduced in the article for the first time.
5. In Figure 2d, the excitation wavelength, monitoring wavelength, and captured delay time should be described in the caption.
6. In "Regulation of enantioselective differentiation and mechanism verification." Paragraph, ".....whereas D-B@L has weak cyan emission with blue afterglow from the host phosphorescence....." D-B@L should be D-Br@L.
7. I cannot find a corresponding chemical structure to the sample code "L-F-H" either in the manuscript or SI.
8. According to Figure S2, the retention time of the D-form and L-form for F-Na appears at approximately 10 min and 15 min, respectively. This is completely different from what Figure S4 displays in the HPLC spectra. However, the authors assigned both of them as the same compound (F-Na). Recheck it!
9. According to this article, the authors only provide absorption spectra in acetonitrile for all title compounds and lack solid-state absorption data. Since either RTP or CPL emission is observed in the solid state, it is necessary to provide solid-state absorption spectra.
10. It seems like the authors prepared this manuscript in a rush, where the organization, statements and labels are not well prepared, which has to be trimmed completely before submission elsewhere.

Author Reply: We first would like to thank all of the three reviewers for their dedicated efforts to improve the manuscript. Based on the critical comments and suggestions, we have performed additional experiments to clarify issues and concerns raised by the reviewers. Given the very long Reply to Reviewers' Comments, we believe that it is perhaps a good idea that, prior to responses to specific comments, we show the reviewers as well as the editor **the logic flow with an emphasis on the mechanistic aspects** of our revised story presented in the Revised Manuscript shown in Figure R1:

Fig. R1 Schematic diagram of the CPE mechanism.

- Observation in fluorescence and phosphorescence intensity disparity.
- Visually observable Chiral-selective Phosphorescence Enhancement (CPE).
- CPE's underlying mechanism investigation, which involves two key points: energy transfer and molecular geometry.
- Schematic diagram of Chirality Dependent Energy Transfer (CDET) and its application in solid-state phosphorescence chiral sensing for naturally occurring amino acids and their synthetic counterparts.

a) **Observation in fluorescence and phosphorescence intensity disparity (i.e., same vs different chirality in the guest-host molecular solid).** We show the initial spectroscopic results of the disparity: under the same experimental conditions, we observed a much higher disparity in phosphorescence intensity (Fig. R1a lower figure) in the solid-state guest-host system compared to fluorescence intensity disparity (Fig. R1a upper figure).

b) **Visually observable Chiral-selective Phosphorescence Enhancement (CPE).** As shown in the photograph, the phosphorescence intensity of the host-guest system with the same chirality (i.e., **0.1% L@L**) was observed to be significantly higher than that of the control group with different chirality (i.e., **0.1% D@L**). However, the enantiomeric enhancement ratio was less pronounced for the fluorescence emission.

- c) **An underlying mechanistic investigation of the CPE phenomenon, which involves two key points: energy transfer and molecular geometry.**

A. Energy transfer. The top figure (Figure R1c) is the solid absorption spectra of the host and guest molecules vs. the wavelength-dependent ep values for the doped samples (w/w = 0.1%). When comparing the two figures, we found that the ep of fluorescence (ep_{FL}) is independent of excitation wavelengths, while the ep of phosphorescence (ep_{RTP}) is positively correlated with the solid-state absorbance of the host material. This indicates that when the excitation light wavelength is < 300 nm, the triplet excited-state energy is initially generated by the host and then transferred to the guest, resulting in a larger ep_{RTP} value for phosphorescence. However, when such an energy transfer process from the host to the guest is no longer possible (*i.e.*, wavelength > 300 nm), the phosphorescence intensity of **0.1% L@L** and **0.1% D@L** are almost indistinguishable, leading to the ep_{RTP} value close to 1. (detailed experiment description is also displayed in Author Reply 1 and 2 for Reviewer 1)

The evidence here is almost irrefutable that triplet-triplet energy transfer, which is absent in fluorescence, is mainly responsible for the observed increased phosphorescence ep value compared to fluorescence. Since phosphorescence energy transfer is limited to the Dexter mechanism while fluorescence energy transfer may occur via both the Dexter and Förster mechanisms, we would then logically deduce that the Dexter mechanism is very likely chirality-dependent. However, a deeper reason beyond the scope of chemistry is possibly involved, which merits further studies from other areas such as quantum physics (e.g., chirality-induced spin selectivity, or CISS which is not well understood).

B. Molecular Geometry. The bottom part showcases the design principle of a modular chemical system containing two moieties: an aromatic component as guest/host phosphor and a chiral analyte readily “clicked together” by an efficient chemical reaction under ambient conditions. Based on the analyses from **A.**, it is clear that different combinations of the enantiomers (different chirality) result in differences in the observed phosphorescence ep values, and therefore, the basis for the recognition application.

- d) **Schematic diagram of Chirality Dependent Energy Transfer (CDET) and its application in solid-state phosphorescence chiral sensing.** We finally show that the schematic diagram of CDET and employed such a pronounced phosphorescence disparity to construct an *in-situ* sensing protocol for naturally occurring amino acids from their synthetic counterparts with a different chirality: the protocol is no more complicated than what life scientists are used to and exhibits the best sensing results in the field of luminescence chiral sensing.

For a paper to be published in a top-tier journal, we usually require one of the two criteria:
A. does this study expand the knowledge scope of our current understanding of a specific

field? B. does this study leads to an application best of its kind? I think the answer in our case is yes to both. We undoubtedly discovered that A) the huge disparity between fluorescence and phosphorescence in chiral sensing is due to triplet-triplet energy transfer for the first time (including our previous study which does not involve fluorescence), and B) this sensing protocol realizes the shortest sensing time with the most variety in analytes within the field of luminescence chiral sensing. Therefore, we are confident that this revised manuscript should possess sufficient novelty and impact for publication in *Nature Communications*.

The following content is our point-by-point responses to specific comments and suggestions made by the reviewers:

Responses to the comments of Reviewer #1:

In this work, the authors developed a new method for the chiral recognition of amino acids by employing room-temperature phosphorescence (RTP). This method demonstrates a remarkable versatility in recognizing over a dozen of different amino acids via distinguishable afterglow at ambient conditions. The mechanism of the recognition has been well studied through elaborate spectral analysis, lifetime analysis, SEM, SFG and control experiments, which clearly clarified the roles of host, guest, and heavy-atom. This work enabled the regulation of enantioselective differentiation ability via adjusting the structure of host or guest molecules and showcased that organic phosphorescence offers comparable or even better sensitivity and potential compared to fluorescence. The manuscript was well-organized and should be of broad interest to the area of solid-state luminescence. Therefore, I recommend its publication in *Nature Communications* after the following minor revisions.

Author Reply: We thank the reviewer for the positive evaluation of our manuscript and recommendation for its publication in *Nature Communications* after minor revisions. It is encouraging to know that the method we developed for chiral recognition of amino acids using room-temperature phosphorescence (RTP) is seen as a significant contribution to the field of solid-state luminescence. We are particularly pleased that the versatility of the method, as well as the comprehensive studies conducted to elucidate the mechanism, have been well received.

Comment 1. In Figure 2b and 2c, why the steady-state photoluminescence (PL) spectra and delayed emission (DE) spectra have different excitation wavelength?

Author Reply 1: We thank the reviewer for raising this important question. The choice of different excitation wavelengths for steady-state photoluminescence (PL) and delayed emission (DE) spectra is to accommodate the specific photophysical behavior observed in the system.

In order to optimize the spectra, we utilized 2D excitation-emission-intensity phosphorescence spectra to investigate the trend of phosphorescence emission intensity with varying excitation wavelengths. (Figure R2a, b) Comparing the spectra for **L@L** and **D@L** samples, we observed a pronounced difference at a shorter excitation wavelength ($\lambda_{\text{ex}} = 247 \text{ nm}$): **0.1% L@L** exhibits stronger phosphorescence, while **0.1% D@L** shows almost no emission. At a longer wavelength ($\lambda_{\text{ex}} = 335 \text{ nm}$), however, the phosphorescence intensities of both samples are nearly identical. The 2D excitation-emission- ep_{RTP} spectrum (Figure R2c, Divide the data in Fig. R2a by the data in Fig. R2b) reveals that as the excitation wavelength increases, the enantiomeric RTP enhancement ratio (ep_{RTP}) decreases. This trend is further confirmed when examining the emission intensity and ep_{RTP} at the peak of phosphorescence ($\lambda_{\text{em}} = 526 \text{ nm}$) (Figure R2d). With increasing wavelengths, the ep_{RTP} value drops from ~ 10 to ~ 1 . We also examined the fluorescence emission intensity trend vs. different excitation wavelengths (Figure R3d). The enantiomeric fluorescence enhancement ratio (ep_{FL}) is independent of absorption wavelengths, fluctuating between 1 and 2. These observations are strong evidence of the triplet-triplet energy transfer and chirality-dependent energy-transfer (CDET) process we have proposed, which serves as the rationale behind our choice of excitation wavelengths for different spectroscopic measurements.

Fig. R2 The 2D excitation-emission-intensity phosphorescence spectra ($\Delta t = 5 \text{ ms}$) of (a) **L@L** and (b) **D@L** in air at 298 K. (c) The 2D excitation-emission- ep_{RTP} graphs ($\Delta t = 5 \text{ ms}$) of **L@L** and **D@L** in air at 298 K. (Divide the data in Fig. R2a by the data in Fig. R2b) (d) Delayed excitation spectra ($\Delta t = 5 \text{ ms}$) of **L@L** and **D@L** ($\lambda_{\text{em}} = 526$

nm) vs. ep_{RTP} values in air at 298 K. (The guest-to-host ratio is 0.1% for all samples in the solid state)

Fig. R3 The 2D excitation-emission-intensity fluorescence spectra of L@L (a) and D@L (b) in air at 298 K. (c) The 2D excitation-emission- ep_{FL} graphs of L@L and D@L in air at 298 K. (Divide the data in Fig. R2a by the data in Fig. R2b) (d) Steady-state excitation spectra of L@L and D@L ($\lambda_{em} = 357$ nm) vs. ep_{RTP} values in air at 298 K. (The guest-to-host ratio is 0.1% for all samples in the solid state)

In the case of the Delayed emission (DE) spectra, as shown in Figure 2c, we aimed to maximize the room-temperature phosphorescence (RTP) intensity, which according to our proposed energy-transfer mechanism, is stronger when the host molecules are excited. To achieve this, we selected an excitation wavelength $\lambda_{ex} = 247$ nm (maximal absorbing wavelength of the host), which indeed yielded both a high intensity of RTP and a high ep value compared to excitations using longer wavelengths than 247 nm.

For the steady-state photoluminescence (PL) spectra, however, using an excitation wavelength of 247 nm would result in a loss of spectral information in the range of 480-560 nm. This interference is due to a significant second-harmonic generation (SHG) peak from the excitation light itself at approximately 494 nm. Besides, the ep_{FL} is independent of absorption wavelengths. Therefore, we opted for an excitation wavelength of 298 nm for the steady-state PL spectra to ensure spectral integrity across both the fluorescence and

phosphorescence ranges. This selection avoids the interference from SHG and allows for a more accurate representation of the photoluminescence properties of our samples.

Comment 2. Since the process of chiral recognition and spectral analysis occurred in the solid state, the authors should provide the solid absorption spectra of host molecules, both pre and post doping host samples are required.

Author Reply 2: We thank the reviewer for the valuable advice. The absorption spectra of the host molecules, the guest molecules and the doped samples in the solid state have been measured and are presented in Figure R4. As illustrated in Figure R4a below, the absorption of host solids occurs in the 200-300 nm range. The absorption of guest molecules occurs in the 200-350 nm range, which extends beyond that of the host. (Figure R4b) When comparing the pre-doping and post-doping host samples, we observed that doping levels below 1% do not significantly alter the solid absorption spectra under 300nm. Meanwhile, the absorption curves of the doped samples between 300-350 nm are similar to those of the guest molecules. Considering that the molar extinction at 247 nm of the host matrix is greater than that of the guest molecule and that the host molecules make up more than 99% of the samples, we can safely conclude that the photon energy at 247 nm is mainly absorbed by the host matrix and then transferred to the guest molecules, while the photon energy at 335 nm is more likely directly absorbed by the guest molecules.

This thorough examination and comparison confirm the integrity of the host structure post-doping and the effectiveness of the doping process at low ratios, which is essential for maintaining the desired uniform photophysical properties of the host material for efficient chiral recognition.

Fig. R4 The solid-state absorption spectra of the host molecules, guest molecules, and doped samples (w/w = 1% or 0.1%).

Incidentally, when comparing the ep_{RTP} values with respect to the solid-state UV absorption spectra of the host and guest molecules, we found that the ep of fluorescence (ep_{FL}) is independent of absorption wavelengths, while the ep of phosphorescence (ep_{RTP}) is positively correlated with the solid-state absorbance of the host material. (Figure R5) This indicates that when the excitation light wavelength is < 300 nm, the triplet excited-state energy is initially generated by the host and then transferred to the guest, resulting in a larger ep_{RTP} value for phosphorescence. However, when such an energy transfer process from the host to the guest is no longer possible (*i.e.*, wavelength > 300 nm), the phosphorescence intensity of **0.1% L@L** and **0.1% D@L** are almost indistinguishable, leading to the ep_{RTP} value close to 1. The evidence is almost irrefutable that the triplet-triplet energy transfer, which is absent in fluorescence, is mainly responsible for the observed phenomenon, verifying the mechanism of CPE theory.

Fig. R5 a) The solid absorption spectra of the host and guest molecules. b) The wavelength-dependent ep values of the doped samples. ($w/w = 0.1\%$)

As to further display our investigation into the mechanism, we have added the following text in “Chiral-selective RTP enhancement for phenylalanine derivatives.” paragraph: “In addition, we compared the 2D excitation-emission-intensity spectra of **L@L** and **D@L** solids (the solid-state absorption spectra are provided in Figure S6), respectively (Figure S12), where it is evident that CDET ceases to exist for the **D@L** sample (Figure S12b showing that RTP intensity diminishes at higher photon energy mainly absorbed by the host solid). The 2D excitation-emission- ep map also supports the conclusion (Figure S12c): ep of RTP is significantly enhanced when the host solid is photo-excited (240-300 nm) rather than direct excitation (300-350 nm) of the guest molecule. When the wavelength-

dependent λ_{em} of RTP is plotted against excitation intensity (Figure S12d), the function coincides with the spectrum of the solid-state absorption of the L host (Figure S13)."

Comment 3. The authors should also provide the scanning electron microscope (SEM) image of doped samples.

Author Reply 3: We thank the reviewer for the important suggestion. We have conducted a Scanning Electron Microscope (SEM) analysis of the doped samples (0.1% w/w) using different processing methods, and have presented the findings in Figure R6. The SEM images provide a detailed view of the morphological characteristics of the host matrix post-doping. These images along with the results of the powder X-ray diffraction (PXRD) (Figure R7) reveal that the samples recrystallized with the same solvent exhibit similar morphologies.

Specifically, the host solids processed by tetrahydrofuran (THF) are organic microcrystals with smaller particle sizes (Figure R6a, b, R7b); the host solids processed by a mixed solvent of chloroform (CHCl_3) and n-hexane (HEX) (90:10) are long belt-like crystals with large aspect ratios (Figure R6c, d, R7b), which could be subject to non-isotropic scattering of the RTP emission and produce inconsistent results. (Figure R6e, f) The data not only confirm the consistency of the doping protocol (with THF) but also provide a clear morphological basis for the observed photophysical property differences. In summary, these SEM images are instrumental in validating our approach and providing a solid foundation for the interpretation of the luminescence data.

We have redrawn the Figure 4a-4f, revised the corresponding caption to exhibit our new findings and added the following texts in "Exploration of morphology and microstructure" Paragraph: "As can be seen from the scanning electron microscope (SEM) images (Figure 4a-4d) and the power X-ray diffraction (PXRD) patterns (Figure S28), doping with guest molecules of different chirality have no observable effect on the microscopic morphologies for solid-state samples obtained from either tetrahydrofuran (THF, Figure 4a-4b) or a mixture of chloroform/n-hexane (CHCl_3 /HEX, Figure 4c-4d). However, the influence of solvent is tremendous: evaporation from the THF solution produces crystals of much smaller sizes with lower aspect ratios, which in contrast to large belt-like crystals acquired from CHCl_3 /HEX. The spectroscopic differences were also measured (Figure 4e-4f), where much stronger RTP intensity and a higher λ_{em} value are achieved for samples obtained from THF."

Fig. R6 SEM images of doped guest-host samples of (a) **L@L** and (b) **D@L** obtained by recrystallization with tetrahydrofuran (THF); (c) **L@L** and (d) **D@L** obtained by recrystallization with chloroform (CHCl₃) and n-hexane (HEX) (90: 10). (e) Delayed emission ($\Delta t = 5$ ms) spectra of **L@L** and **D@L** by recrystallization with THF (solid borderline) and CHCl₃: HEX = 90: 10 (dash borderline) in air at 298 K ($\lambda_{\text{ex}} = 247$ nm). (f) The phosphorescence intensity of different doped samples for different solvent systems. (The guest-to-host ratio is 0.1% for all samples in the solid state)

Fig. R7 Powder X-ray diffraction (PXRD) patterns of doped samples at $w/w = 1\%$ or 0.1% for different solvent systems.

Comment 4. How the enantiomer excess (ee) values of the chiral analytes (amino acids) affect the ratios of chiral recognition? The authors could provide at least one example.

Author Reply 4: We thank the reviewer for raising the important question regarding the impact of enantiomer excess (e.e.) values on the ratios of chiral recognition. We have thoroughly investigated this relationship using phenylalanine as an exemplar chiral analyte. The results are presented in Figure R8a, and the key findings are summarized in Figure R8b.

As demonstrated in Figure R8, the photoluminescence intensity at 526 nm decreases systematically with increasing enantiomeric excess of the D-enantiomer, indicating a direct correlation between the e.e. value and the efficacy of chiral recognition. This trend is observed across a broad range of e.e. values, from a racemic mixture (50:50) to nearly enantiopure samples. It is worth noting that the phosphorescence intensity diminished rapidly when a small amount of the D-enantiomer was doped in the L-host, showing the sensitivity of the host matrix to trace enantiomers.

These observations are pivotal as they validate our proposed mechanism where the chiral recognition process is sensitive to the stereochemical composition of the analyte. This sensitivity to e.e. values showcases the potential of this RTP-based method for the determination of enantiomeric composition in chiral mixtures, which is of significant interest

for applications in enantioselective synthesis and chiral resolution processes. We have added the Fig. R8 into the Supporting Information as Figure S57.

Fig. R8 The RTP emission intensity at 526 nm and correlation between the enantiomer excess (e.e.) values, illustrating the relationship between e.e. values and the efficiency of chiral recognition.

Comment 5. In rapid test protocol section, the characterization lacks details: what is the power of UV lamp, the distance from samples, and excitation time for the delayed emission or afterglow?

Author Reply 5: We thank the reviewer for the suggestion to provide more details in our rapid test protocol section regarding the characterization parameters. We have now included the detailed methodology and characterization for the rapid test protocol in the revised Supporting Information. The specific parameters used in our experiments are as follows: The excitation light source is a hand-held UV lamp emitting at 254 nm with an input power of 6 watts. The distance between the lamp and the samples was maintained at ~10 centimeters, and the excitation time for obtaining delayed emission or afterglow in the samples was set to 10 seconds (long excitation time such as 10 s is not necessary but is easily operable for generating a consistent protocol). These parameters were carefully optimized to ensure consistent and reproducible results across all measurements, and we believe that the inclusion of these details will greatly enhance the clarity and reproducibility of our protocol.

Comment 6. There are few formatting errors in manuscript and supporting information. For example: the abbreviation “CDET” needs to be defined when it first appears in the main text. In “Development of a rapid test protocol for amino acids” section, “0.5ml” and “2M HCl”, there should be a space in front of the unit. The same problem also appeared in the supporting information.

Author Reply 6: We thank the reviewer for catching these mistakes and we have fixed them accordingly in the manuscript.

Responses to the comments of Reviewer #2:

Zhang et al. provided a universal design strategy to construct an amino acids-based chiral guest-host RTP system. The system has photoluminescence enhanced by chiral selective phosphorescence in the solid state. This exceptional property can quickly identify the absolute configuration of amino acids. I believe the protocol is novel and interesting, and the characterization is extremely adequate, but the authors should pay attention to the following issues before the publication in Nat. Commun.

Author Reply: We greatly appreciate the positive comments from the reviewer on our universal design strategy and the recognition of the novelty and interest of our protocol for the rapid identification of the absolute configuration of natural amino acids through enhanced photoluminescence by chiral selective phosphorescence (CPE) in the solid state. We are delighted with the reviewer's assessment of our characterization as extremely adequate.

Comment 1. The authors claimed the protocol is rapid compared to other methodologies, which is determined by the reaction speed. Is there any possibility to further accelerate the reaction by adding some catalyst or reagents or modifying reaction conditions?

Author Reply 1: We thank the reviewer for the valuable advice. During the protocol development process, we have attempted to optimize the reaction conditions by changing the reagents such as the base (e.g., triethylamine or sodium bicarbonate) and the solvent (e.g., dichloromethane or dioxane: H₂O=3:1), and these optimizations have reduced the reaction time from hours to a few minutes, we found the optimal reaction condition is sodium hydroxide as base and tetrahydrofuran as solvent, which is presented as a standard protocol in SI.

According to the reviewer's kind suggestions, we have tried to further accelerate the reaction by adding catalyst/reagents like 4-dimethylaminopyridine or 18-Crown-6. We found the addition of 1.0 eq. 18-crown-6 to potassium hydroxide could reduce the reaction time to a few seconds for phenylalanine with almost no influence on the enantiomeric RTP enhancement ratio (e_{RTP}).

We also believe that automated operations or the combination of AI based on the existing protocol could further accelerate reactions and optimize the chiral discrimination of amino acids, and we are now conducting relevant research with our collaborators.

Comment 2. Is there any explanation for the unusual PL spectra of F-Ph-DL solid, F-Ph-D solid, and F-Ph-L solid in Figures S7d and S7f?

Author Reply 2: We appreciate the reviewer's inquiry into the unusual photoluminescence (PL) spectra observed for **F-Ph-DL** solid, **F-Ph-D** solid, and **F-Ph-L** solid in Figures S7d and S7f.

The unusual PL spectra observed can be attributed, according to Kasha's exciton theory, to the slight variation of molecular arrangements and packing in the solid state, which is influenced by temperature, air humidity, the speed of solvent evaporation, and the speed of crystallization. In addition, the highly scattered nature of microcrystalline samples may also contribute to such variations. We carefully remeasured the emission spectra (Figure R9) and noted that indeed, the anomalous behaviors merely differ in vibronic structures, consistent with the Kasha's exciton theory that small differences in packing usually induce a difference in intensity ratios between the 0-0 band and other bands. (Spano et al., Chem. Rev. 2018, 118, 7069–7163). Incidentally, it is not uncommon to see different solid-state PL spectra even for the same sample based on our past experiences. Since we compared the difference of intensity by choosing doping chiral isomers with the same host matrix, (e.g., **L@L** and **D@L**) the crystalline properties of the host solid did not affect the conclusion of the CPE phenomenon.

Fig. R9 a) Normalized photoluminescence spectra at 77 K. b) Normalized delayed emission ($\Delta t = 5$ ms) spectra at 77 K. c) Powder X-ray diffraction patterns of the host solid.

Comment 4. The author believes that the energy transfer between the host and the guest is carried out through the “Dexter” mechanism, why is “Dexter” not “Förster”?

Author Reply 4: To facilitate a clearer and more logical understanding, we propose addressing **Comment 4** prior to **Comment 3**.

We thank the reviewer for the great question, and this is a critical question and very important for understanding the mechanism, we try our best to address the reviewer's concern. The Dexter energy transfer mechanism is characterized by its ability to facilitate energy transfer through electron exchange, which can occur in both singlet and triplet excited states. This contrasts with the Förster mechanism, which is limited to energy transfer in singlet states via dipole-dipole interactions, as pure triplets commonly found in organic molecules have near zero excited state dipole moment. (Turro, Nicholas J. *Modern molecular photochemistry*. University science books, 1991.)

If the energy transfer in our system were to occur via the same mechanism irrespective of spin multiplicities (Namely Förster type), we would expect the fluorescence (F) and phosphorescence (P) from the guest molecules to have the same λ_{em} values. However, our experimental results, specifically shown in Figures 2b, 2c, and 3c, demonstrate that the λ_{em} values for fluorescence range from 1.9 to 3.2, while the λ_{em} values for RTP are significantly higher, ranging from 6.0 to 9.3. Besides, we observed distinctly different patterns for F and P in the 2D excitation-emission-intensity phosphorescence spectra, respectively, which are usually indications of different energy-transfer pathways. (Figure R2 and R3) The newly supplemented figure shows that the λ_{em} of fluorescence ($\lambda_{em,FL}$) is independent of absorption wavelengths, while the λ_{em} of phosphorescence ($\lambda_{em,RTP}$) is positively correlated with the solid-state absorbance of the host material. (Figure R5) This observation indicates that fluorescence and phosphorescence are likely arising from different energy-transfer sources.

Since that 1) both the Förster and Dexter mechanisms are allowed in F and only the Dexter mechanism is allowed for P, AND 2) that the energy transfer is indeed observed spectroscopically, it is only logical to deduce that energy transfer in P (i.e., Dexter) must be responsible for the observed high disparity while F low disparity. **In conclusion, the main contribution to the λ_{em} values of RTP arises from Dexter-type triplet-triplet energy transfer.**

As for the possible confusion of the energy transfer mechanism, we have added the following texts in the "Influences of different guest-to-host ratios in solid-state samples." paragraph: "The disparity in λ_{em} values between fluorescence and RTP is attributed to the fact that guest fluorescence can occur via both Förster and Dexter types of energy transfer while guest RTP is only limited to the latter type. We then obtained wavelength-dependent λ_{em} values for fluorescence emission as well, and found that these values are independent of excitation energy (Figure S19), clearly indicating that the long-ranged dipole-dipole Förster process is not sensitive to chirality. Based on these experiments results, we can then deduce that the shorter-ranged Dexter energy transfer, which is the sole energy transfer mode for RTP, is responsible for such enhanced λ_{em} differences."

Although not mentioned in the manuscript, we would like to emphasize that in this specific benzene-naphthalene energy transfer system, the contribution of the Förster-type is expected to be minimal since the oscillator strength of the benzene $S_1 \rightarrow S_0$ ($\tilde{A}_1B_{2u} \rightarrow \tilde{X}_1A_{1g}$)

transition is extremely small due to orbital symmetries, which is easily outcompeted by electron exchange (Dexter type) in the solid state. However, substitutions can significantly increase the allowedness because of vibronic coupling to the optically allowed \check{C}_1E_{1u} state.

Comment 3. Please further explain the disparity in ep values between fluorescence and RTP. I do not receive the critical point using the Dexter-type energy transfer model. Also, the disparity in ep values between L@L and D@D cases is wondering.

Author Reply 3: We thank the reviewer for the kind question. The reason for “the disparity in ep values between fluorescence and RTP” is presented in **Author Reply 4** above. To reiterate, the source of phosphorescence in our system is predominantly from the triplet-triplet energy transfer pathway, which is characteristic of the Dexter-type energy transfer. This differs from fluorescence, which can involve both Dexter- and Förster-types.

Regarding the disparity in ep values between L@L and D@D cases, this can be attributed to the intrinsic variability and semi-quantitative nature of luminescence spectroscopy, particularly in the solid state. In order to address this variability and provide robust data, we have conducted multiple measurements (at least 3 times) for each individually prepared sample set and re-calculated the ep values. (Figure R10) As shown in Table R1., The re-calculated average ep values for the L@L and D@D cases provide a more accurate representation of the luminescence characteristics and the efficiency of chiral recognition in our system.

Fig. R10 Steady-state photoluminescence (PL) spectra of two chiral guests ($w/w = 0.1\%$) in a) the L-host solid and b) the D-host solid at 298 K ($\lambda_{\text{ex}} = 298$ nm). (The number following the sample name indicates the different individually prepared samples.) Delayed emission ($\Delta t = 5$ ms) spectra of two chiral guests ($w/w = 0.1\%$) in c) the L-host solid and d) the D-host solid at 298 K ($\lambda_{\text{ex}} = 247$ nm).

Table R1. Averaged ep values of dopant samples ($w/w = 0.1\%$) at room temperature.

	0.1% L@L / 0.1% D@L	0.1% D%D / 0.1% L@D
$ep_{\text{FL}}^{\text{a}}$	1.6	1.9
$ep_{\text{RTP}}^{\text{a}}$	9.3	9.5

^a The average result of 3 independent trial processes.

We have updated the ep values of dopant samples ($w/w = 0.1\%$) in the main text to reflect more accurate results and reduce possible confusion.

Comment 5. Why is energy transfer better in the same configuration? What is the internal relationship between energy transfer and chirality?

Author Reply 5: We thank the reviewer for this core question.

In our previous responses (**Author Reply 3, 4**), we have discussed the role of chirality-dependent energy transfer (CDET) in the efficiency of the ep values in RTP. The efficiency of energy transfer in configurations where the host and guest share the same chirality

(referred to by the reviewer as “the same configuration”) can indeed be higher, but the fundamental reasons for CDET are believed to be complex and multifaceted.

To validate the CDET mechanism, we first investigated whether chirality and energy transfer are necessary factors in the CPE phenomenon. Therefore, we chose polymethyl methacrylate (PMMA) and polystyrene (PS) as the host matrix to exclude the influence of chirality energy transfer between host and guest. As shown in Figure R11, in the absence of chirality and energy transfer, there is little difference in RTP intensity between different chiral guests, which supports our hypothesis in the Schematic diagram of the CPE mechanism. (Figure R1)

Fig. R11 a), b) Delayed emission ($\Delta t = 5$ ms) spectra of the polymethyl methacrylate (PMMA) and two chiral guests (w/w = 1% or 0.1%) in PMMA in vacuo at 298 K ($\lambda_{\text{ex}} = 247$ nm). c), d) Delayed emission ($\Delta t = 5$ ms) spectra of the polystyrene (PS) and two chiral guests (w/w = 1% or 0.1%) in PS in vacuo at 298 K ($\lambda_{\text{ex}} = 247$ nm).

To explore the internal relationship between chirality and energy transfer, we also compared the influences of non-radiative transition rate (k_{nr}) and exciton quenching rate (k_{q}). We measured the quantum yields of fluorescence and phosphorescence of the doped samples, as well as the temperature-dependent lifetimes within a range of 80 K to 320 K. (figure R12) The lifetime value as a function of temperature could be fitted using a sum of two exponential functions. (S. Hirata, *Adv. Sci.* **2019**, *6*, 1900410) The photophysical parameters of the doped samples are listed in Table R2. We can conclude that k_{nr} and k_{q}

are not the main reasons for the conspicuous difference in τ_p values between different samples (especially at low temperatures).

Fig. R12 Temperature dependence of the phosphorescence lifetime of doped samples ($w/w = 1\%$ or 0.1%).

Table R2. Photophysical parameters of the doped samples.

Sample	τ_f (RT) (ns)	ϕ_f (%)	τ_p (RT) (s)	ϕ_p (%)	k_f^a (s^{-1})	k_p^b (s^{-1})	k_{nr}^c (RT) (s^{-1})	k_q^c (RT) (s^{-1})	k_{nr}^c (77 K) (s^{-1})	k_q^c (77 K) (s^{-1})
1% L@L	10.6	38.95%	0.581	1.25%	$3.67E+07$	$3.53E-02$	$6.96E-01$	$3.13E-01$	$6.33E-01$	$6.42E-08$
1% D@L	7.53	20.85%	0.322	0.65%	$2.77E+07$	$2.56E-02$	$7.17E-01$	$2.23E+00$	$7.05E-01$	$3.48E-23$
0.1% L@L	7.65	36.17%	0.637	2.93%	$4.73E+07$	$7.21E-02$	$6.69E-01$	$2.58E-01$	$5.77E-01$	$2.77E-09$
0.1% D@L	5.09	26.20%	0.322	0.90%	$5.15E+07$	$3.77E-02$	$7.87E-01$	$7.62E-01$	$6.53E-01$	$4.63E-10$

$$^a k_f = \phi_f / \tau_f$$

$$^b k_p = \phi_p / [(1-\phi_f) \times \tau_p]$$

$$^c k_p + k_{nr} + k_q = 1 / \tau_p$$

Based on the above discussion, the first reason to consider is the (much more sensitive) Dexter energy transfer rate's dependence on distance compared to that of the Förster energy transfer rate. Dexter energy transfer has a short-range mechanism that requires the orbital overlap of the donor and the acceptor. As equation (1) suggests, the rate of energy transfer decreases exponentially with the increase in distance (R) between the donor and the acceptor molecules.

$$k_{ET}(\text{Dexter}) = KJ \exp\left(\frac{-2R}{L}\right) \quad (1)$$

To probe the average distance between doped guest and host molecules, we analyzed the results from the sum frequency generation (SFG) spectroscopy. The orientation angle of the phenyl group (θ_{ph}) can be deduced in terms of the $\chi_{ppp}^{(2)}/\chi_{ssp}^{(2)}$ ratio of the ν_2 mode. Under the assumption of a δ -distribution, the average orientation angles are determined to have

little change among different doped samples. (Figure R13 a) With the knowledge of the orientational information, we estimate the relative structural ordering (RSO) of the doped samples. When the doped host-guest molecules have the same chirality, the RSO of the doped system is a little higher than that of the chiral isomer (Figure R13 b). Therefore, in systems where the host and the guest share the same chirality, the SFG spectra suggest that there might be a more favorable molecular packing. This allows closer proximity and better orbital overlap, thereby facilitating more efficient Dexter energy transfer, the predominant energy-transfer pathway as has been discussed multiple times.

Fig. R13 (a) The relationship between the $\chi_{pppp}^{(2)}/\chi_{sspp}^{(2)}$ ratio and θ_{ph} . (b) The relative structural ordering (RSO) of the doped samples.

However, it would be an over-stretch to claim that such large ep values are the result of such minor differences in photophysical parameters or molecular configuration of the doped samples, especially when little differences were observed for the 0.1% doped samples from SFG spectra.

From the abovementioned analyses, we speculate that there might be deeper and more intricate reasons beyond the scope of regular characterization methods in chemistry. It must be emphasized that these reasons are part of our (long-term) ongoing research. For example, it could be related to a process called chirality-induced spin selectivity (CISS), where the electron hopping is sensitive to the chirality of a molecule as has been recently demonstrated by Wasielewski et al. (Wasielewski et al. *Science* **382**,197-201(2023).) Such mechanisms would definitely warrant investigation at the single-molecule level and may take years to uncover.

In conclusion, while we have exhausted all potential reasons within our ability for the enhanced efficiency of RTP intensity in homochiral systems, further research is essential to definitively determine the “internal relationship” between energy transfer and chirality. We are actively engaged in this exploration and look forward to contributing further to the understanding of this intricate interplay.

Comment 6. Why does the 0.1% ratio give the best ep value?

Author Reply 6: We appreciate the reviewer's question regarding the optimal doping ratio for achieving the best ep value.

The doping ratio plays a crucial role in determining the efficiency of the chiral recognition process. At higher ratios, such as 1%, there is an increased likelihood of direct excitation of the guest molecules leading to RTP, which can compromise the specificity of the chiral recognition. Additionally, at these higher ratios, the guest molecules may become more prone to self-absorption and reabsorption processes that can affect the luminescence properties.

Conversely, at ratios lower than 0.1%, the intensity of the RTP from the homochiral pairings is reduced, resulting in a lower signal-to-noise ratio. This makes it challenging to distinguish between the luminescence signals from homochiral and heterochiral pairings accurately.

Moreover, the efficiency of triplet exciton migration, which is essential for energy transfer leading to RTP, is limited by distance. At ratios below 0.1%, even if the chirality of the host and guest are matched, the exciton may not efficiently transfer energy due to the increased separation between the host exciton loci and guest molecules. This insufficient energy transfer can result in lower ep values, as the ep value reflects the ratio of luminescence intensity between homochiral and heterochiral host-guest pairs.

Therefore, a ratio of 0.1% represents an optimal balance where the RTP signal is strong enough to yield a high signal-to-noise ratio without the complications introduced at higher ratios. At this ratio, the exciton migration is efficient within homochiral systems without potential problems with too many guest molecules, enhancing the discriminatory power of the RTP process and resulting in higher ep values.

Comment 7. Why does L@L observe chiral-selective phosphorescence enhancement, while D@DL does not follow the same phenomenon, as the racemic host also contains the L configuration?

Author Reply 7: We thank the reviewer for their astute question regarding the selective enhancement of phosphorescence in chiral systems.

In the case of the **D@DL** sample, the photoluminescence properties are indeed a composite of both the **D@L** and **D@D** interactions. Similarly, the **L@DL** sample displays photoluminescence characteristics that are a combination of **L@D** and **L@L** interactions. The crux of the phenomenon lies in the chiral-selective phosphorescence enhancement (CPE), which is contingent on the specific chirality pairing of the host and guest.

Phosphorescence enhancement is observed when the chirality of the host and guest are the same, hence the **L@L** pairing shows this enhancement. On the other hand, the **D@D**

pairing, by the same principle, would also exhibit similar phosphorescence enhancement due to the matched chirality.

When we consider the **D@DL** sample, which contains a racemic host, we see that the enhancement does not occur to the same extent as in the homochiral pairings. This is because the **D@DL** sample has an equal mixture of both enantiomers of the host, and thus the interaction with the guest is not exclusively homochiral. While the D enantiomer of the guest will interact with the D enantiomer of the host to produce some degree of enhancement, the L enantiomer of the host is also present and will interact with the D guest, but without the same degree of enhancement effect. The result is an averaged outcome that does not exhibit the same level of chiral-selective phosphorescence enhancement observed in homochiral systems.

In conclusion, the racemic host in **D@DL** does not *a priori* exhibit the same CPE phenomenon because the enhancement is diluted by the presence of both chiral interactions, *i.e.*, those that can lead to enhancement (**D@D**) and those that cannot (**D@L**), resulting in a non-selective overall phosphorescence response. This underscores the importance of chiral purity in the host for achieving the best CPE results.

Comment 8. To demonstrate that doping brings about changes in energy transfer rather than suppression of non-radiation. The author could give the k_{nr} and k_{ISC} of **D@D**, **D@L**, and **D@D**.

Author Reply 8: We thank the reviewer for the suggestion to further elucidate the impact of doping on energy transfer processes, rather than on the suppression of non-radiative decay mechanisms.

As indicated in Figure R12, we have conducted measurements of the phosphorescence lifetime (τ_p) of the **L@L** and **D@L** doped samples across a temperature range from 80 K to 320 K.

The results, which are summarized in Table R2, show that the non-radiative decay constants (k_{nr}) for different samples exhibit minimal variation, suggesting that the intrinsic non-radiative decay processes are not significantly altered by doping under these experimental conditions. This is an important observation as it indicates that the doping process itself does not fundamentally suppress or enhance non-radiative decay, which is not surprising since the lattices of the solid-state crystalline medium are not expected to change over this temperature range.

As for k_{ISC} , it only becomes a deterministic factor when 1) its rate is significantly lower than that of the triplet-triplet energy transfer process and 2) there is a significant difference in the inherent ISC rates for the host molecules in the **L@L** and **D@L** samples. Clearly, the current experiment results do not support this hypothesis, as ISC is not expected to be sensitive to enantiomeric configurations.

In conclusion, this piece of data supports the claim that the observed changes in photophysical behavior upon doping are primarily due to modifications in energy transfer mechanisms that are sensitive to the chiral configuration of the system, rather than being attributed to a general suppression of non-radiative decay pathways. The distinction between homochiral and heterochiral pairings in terms of their quenching efficiencies underscores the role of chirality in dictating the efficiency of energy transfer in these systems.

Comment 9. The manuscript needs to be carefully examined as well as ESI. There are several corrections that need to be made, such as “Deter energy transfer requires...” → “Dexter energy transfer requires...”, “naonolLED-280...” → “nanolLED-280...”, “whereas D-B@L has weak...” → “whereas D-Br@L has weak...”

Author Reply 9: We thank the reviewer for pointing out these issues and they have been fixed accordingly.

Responses to the comments of Reviewer #3:

The manuscript entitled “Rapid Room-Temperature Phosphorescence Chiral Recognition of Natural Amino Acids” provided by Zhang et al. reports a series of phenylalanine-based CPL/RTP materials. The F-Ph enantiomer (host) and F-Na enantiomer(guest) are synthesized by incorporating the original L-/D-phenylalanine moiety with phenyl unit and naphthalene, respectively. The authors emphasize that the designed host molecule (F-Ph –L) can modulate the RTP emission intensity of different chiral guest molecules (F-Na –L and F-Na –D) through the chiral catalytic triplet-triplet energy transfer. Similar studies on the generation of chiral-selective RTP based on energy transfer between host and guest have been reported by the authors’ previous work (Nat Commun 14, 1514 (2023)). **Accordingly, this article appears to be less innovative, which relies on similar concept for applications, while the underlying “mechanism” remains somewhat elusive.**

Author Reply: We thank the reviewer for the feedback on our manuscript. We appreciate the opportunity to clarify the novel aspects of our work and address concerns regarding the perceived innovation relative to our previous report in *Nature Communications*.

This work stands apart from our prior publication in several key areas that represent the state-of-the-art:

i) **Real-time ambient-condition photoluminescence chiral sensing:** Unlike the previous work, which required harsh reaction conditions and rigorous purification of chiral analytes impractical for real applications, this study achieves rapid, real-time chiral recognition via *in-situ* reactions under ambient conditions similar to a commercial protocol used by biologists. The emphasis on real-time sensing is a significant leap forward, enabled by the introduction of acetyl chlorides to enhance reaction efficiency, which diverges from the structural complementarity focus of our earlier work.

ii) **Universality of the CPE phenomenon and validation of the CDET mechanism:** While the prior study introduced the CPE phenomenon within a specific PI/NI system, its universality was not established. The current work broadens the applicability of CPE using a distinctly different system and confirms the reliability of the CDET mechanism through the most common chiral substance: amino acids. Besides, we investigated the underlying mechanism of CEDT which was previously only hypothesized by offering the irrefutable evidence of triplet-triplet energy transfer. Furthermore, the CDET mechanism is established by comparing the ep value disparities for fluorescence and phosphorescence, which was absent in the previous study.

iii) **Regulated sensing parity for L and D isomers:** The ability to modulate the sensing performance between L and D enantiomers enhances the credibility of the CDET mechanism and represents a novel regulatory experiment within the field of chiral recognition.

iv) **Advanced applications:** We present the most efficient photoluminescence chiral recognition of amino acids using a single molecular-solid sensor to date, characterized by the shortest sensing time, the most extensive variety of amino acid substrates, and the lowest sensing concentration reported. In addition, we optimized the experimental scheme by exploring the morphology, found the sensitivity of the detection method to trace chiral impurities by doping different proportions of chiral objects, and achieved a higher degree of differentiation by molecular modification. Furthermore, we also investigated the potential CPL properties of these chiral doping systems.

Fig. R14 CPL emission spectra of doped samples ($w/w = 0.1\%$) excited at 298 nm. The DC value in the bottom spectrum stands for luminescence intensity.

Regarding the mechanism, we have provided a more detailed and robust explanation to address the questions raised. As shown in Figure R1, we found two key factors that determine the CPE phenomenon: the molecular geometry of the host and the guest, and the triplet-triplet energy transfer between the host and the guest. The hypothesis was improved by comparing the solid absorption spectra of the host and guest molecules along with the wavelength-dependent ϵ_{RTP} values of the doped samples ($w/w = 0.1\%$). (Figure R5.) The positive correlation of the ϵ_{RTP} with the solid-state absorption of the host material is solid evidence that triplet-triplet energy transfer, which is absent in fluorescence, is mainly responsible for the CPE phenomenon. As highlighted in response to Reviewer 2's comments, the choice of the Dexter mechanism as the operative energy transfer process over the Förster mechanism is a pivotal aspect of our findings. Our discussions in Author

Reply 4 and Author Reply 3 outline the rationale behind this choice and its importance in the observed CPE phenomenon. In the current manuscript, additional evidence such as substituents-sensitive CDET behaviors, and variable-temperature phosphorescence lifetime is also supplied to further corroborate the mechanism. The detailed findings and discussions pertaining to these measurements and other concerns raised by the reviewers are presented in subsequent author replies.

We trust that the revisions and additional data presented will underscore the innovative nature of our work and its marked distinction from the previous study – not a routine continuation but a meaningful step in the field.

Regarding the chiral-selective amino acid recognition, what is puzzling me is why the RTP photoluminescence obtained by doping different types of chiral amino acid molecules (guests) using the same host material indeed obtained the same results. In other words, why all D/L amino acids follow the same trend whereas I do not see any “common” interaction!

Author Reply: We thank the reviewer for this insightful question.

It is indeed intriguing that energy donors of various chiral amino-acid structures, when doped into the same host material, exhibit a consistent trend in room-temperature phosphorescence (RTP) photoluminescence. This observation suggests a more fundamental mechanism at play, one that is not solely dependent on specific molecular interactions, as has been observed in conventional luminescence chiral sensing, but rather on a commonality shared by the chiral component of the amino acids.

Although not completely understood, the consistent RTP behavior across different chiral amino acids may be attributed to two key factors:

- i) Modification by the naphthoyl group (Na): All the amino acids in our study have been modified by incorporating a Na group. This group serves as the RTP phosphorescence, which primarily influences the spectral properties, whereas the amino acid itself acts as the chiral center that determines the chiral-selective phosphorescence enhancement (CPE) effect.
- ii) Intrinsic Properties of Chirality: Our results suggest that the observed CPE phenomenon does not stem from structural complementarity or specific intermolecular interactions between the host and guest molecules. Instead, it is based on the intrinsic properties of chirality. The similarity of chirality between host and guest appears to dictate the RTP photoluminescence outcome. This led us to propose the hypothesis of chirality-dependent energy transfer (CDET) in the binary system.

As has been stated above, the underlying factors influencing CDET are likely multifaceted. They could include the interaction differences of excited state dipole moments between the host and guest, and the distance dependence of the Dexter energy transfer rate, as mentioned. The similarity in chirality may facilitate more efficient energy transfer due to better orbital overlap, preferred phase configuration for triplet exciton, or more effective electron hopping as described by CISS, which are all favorable for Dexter-type (two-step electron hopping) energy transfer. We have provided detailed discussions in Reply 4 in Reviewer 2's Comments. However, validating these hypotheses requires substantial technical diligence. Currently, we have made a few plans to use more advanced facilities including synchrotron, free-electron laser, and various high energy particle sources to study the systems.

Regardless, this commonality in the behavior of different chiral amino acids underscores the universal nature of the CDET mechanism and its potential applicability across a wide range of chiral recognition scenarios. Our study has thus uncovered a fundamental aspect of chirality that governs the RTP photoluminescence in solid-state molecular systems, independent of the specific amino acid being used.

In my opinion, this manuscript lacks comprehensive explanations of the mechanism, which gives the impression of a collection of observations and perhaps potential applications without a solid scientific foundation. Given these considerations, I am not convinced of its suitability for publication in *Nature Communications*.

Author Reply: We sincerely regret that our manuscript gave the impression of lacking a comprehensive explanation of the underlying mechanism. We understand the importance of a solid scientific foundation, particularly for a journal of *Nature Communications* caliber, and we appreciate the opportunity to clarify our approach.

We believe that the proposed mechanism, although complex, provides a viable scientific basis for the chiral recognition of amino acids and potentially other chiral substances. The mechanism is rooted in the interplay between chirality and photophysics, specifically through CDET mechanism, which we have substantiated with various experiments.

In response to the concerns raised:

The universality of the CDET Mechanism: We have found the CDET mechanism by examining the interactions between host and guest molecules (i.e., the specific PI/NI system) in the previous publication. In the current study, we have been able to extend the theory into other molecular systems and for the first time demonstrated the universality of CDET.

New Experimental Evidence in the Current Study: The observations reported here are supported by comprehensive experimental data, including the sum frequency generation

(SFG) spectra, 2D photoluminescence spectra, and temperature-dependent lifetime measurements. These experiments collectively validate our hypothesis on CDET.

Practical Applications: While we have indeed discussed potential applications of our findings, we have ensured that these discussions are grounded in the empirical data and theoretical understanding presented in our work.

In light of these points, we have provided new knowledge beyond current understanding and a new application that is the best of its kind. We hope that the revised manuscript will address the concerns and establish the suitability of our work for publication in Nature Communications.

Other major concerns:

Comment 1. Upon reviewing Figure S7, for F-Ph-L and F-H-D, I observed the presence of an emission band within the 370-550 nm range at 77 K (Fig. S7 c-d). Moreover, there is also a delayed (5 ms) emission band within the 350-500 nm range, which is blue shifted from the previous emission at 77 K (Fig. S7 e-f). Explain.

Author Reply 1: We thank the reviewer for raising this problem. The confusion arises from the fact that the two horizontal axes are not on the same scale, which gives an illusion of blue-shifting if not careful. To avoid potential confusion for readers, we have now replotted the Figures using a uniform scale range on the horizontal axis. As for the slight variation in spectra, we have explained in detailed for Reviewer 2 in Reply 2.

Fig. R15 a) and b) Photoluminescence (PL) spectra at 298 K, solution: in dimethyl tetrahydrofuran (2-Me-THF) (2 mM) c) and d) PL spectra at 77 K, solution: in 2-Me-THF (0.04 mM) e) and f) Delayed ($\Delta t = 5$ ms) emission at 77 K, solution: in 2-Me-THF (0.04 mM)

Comment 2. The title of this manuscript is “Rapid Room-Temperature Phosphorescence Chiral Recognition of Natural Amino Acids”. Additionally, at the beginning of the manuscript, the authors stated, “...To realize rapid and reliable sensing, the benzamide molecule is processed into nanocrystals by lyophilization from a mixed solvent, which allows for efficient triplet-triplet energy transfer to the chiral analytes generated in situ from chiral amino acids.” It appears that rapid RTP is one of the key issues addressed in this paper, and the morphology strongly influences the efficiency of energy transfer rates. However, the characterization of the material’s microstructure is rather limited. For instance, SEM cannot identify crystal structure. Other appropriate measurements should be applied to support the mechanism proposed by the authors. Furthermore, to my knowledge,

lyophilization is not necessarily used to create crystals. The experimental parameters of the lyophilization process also significantly affect the material's morphology. These parameters are critical and should be described in detail.

Author Reply 2: We acknowledge the reviewer's concerns about the microstructure characterization of our materials and its potential impact on the energy transfer efficiency crucial for rapid room-temperature phosphorescence (RTP) chiral recognition.

We have indeed utilized Scanning Electron Microscopy (SEM) to visualize the morphology of the nanocrystals obtained via different preparation methods. While SEM provides valuable information about the morphology, we concur that it does not reveal the crystal structure. To address this, we have complemented our SEM analysis with powder X-ray diffraction (PXRD) measurements, which offer comparative insights into the crystallographic structure of different samples. As shown in Figure R16, the samples obtained by the same process have similar morphology.

Fig. R16 SEM images of doped guest-host samples of (a) **L@L** and (b) **D@L** obtained by lyophilization with tetrahydrofuran (THF) and H₂O (90: 10); (c) **L@L** and (d) **D@L** obtained by recrystallization with chloroform (CHCl₃) and n-hexane (HEX) (90: 10). (The guest-to-host ratio is 0.1% for all samples in the solid state). PXRD patterns of doped samples at (e) w/w = 1% and (f) w/w = 0.1% for different solvent systems.

Considering that a rapid test protocol should not involve complex operations such as lyophilization, we attempted to prepare and characterize the sample by directly recrystallizing it with tetrahydrofuran (THF). To our delight, the sample obtained by direct

recrystallization with THF showed a high degree of consistency with the sample obtained by lyophilization with tetrahydrofuran (THF) and H₂O (90: 10), as characterized by SEM and PXRD. (Figure R17)

Fig. R17 SEM images of doped guest-host samples of (a) L@L and (b) D@L obtained by lyophilization with tetrahydrofuran (THF) and H₂O (90: 10); (c) L@L and (d) D@L obtained by recrystallization with tetrahydrofuran (THF). (The guest-to-host ratio is 0.1% for all samples in the solid state). PXRD patterns of doped samples at (e) w/w = 1% and (f) w/w = 0.1% for different solvent systems.

In summary, samples prepared by lyophilization from a mixed solvent of tetrahydrofuran (THF) and water, and those obtained by direct recrystallization with THF are similar to organic microcrystals with smaller sizes. In contrast, samples processed by a mixed solvent of chloroform (CHCl_3) and n-hexane (HEX) (90:10) are long belt-like crystals with large aspect ratios, which correlates with their reduced and even poor performance in chiral recognition. (Figure R6e, f) These observations support the importance of crystal morphology and size in the efficiency of RTP chiral sensing. Based on the concept of rapid and efficient identification, we optimized the process used in this protocol as preparing the sample by directly recrystallizing with THF. It should be clarified that the preparation method of the sample only affects the value of ep and does not change the main conclusion that there is CPE phenomenon between chiral hosts and guests due to the CDET mechanism.

Fig. R6 SEM images of doped guest-host samples of (a) L@L and (b) D@L obtained by recrystallization with tetrahydrofuran (THF); (c) L@L and (d) D@L obtained by recrystallization with chloroform (CHCl_3) and n-hexane (HEX) (90: 10). (e) Delayed emission ($\Delta t = 5$ ms) spectra of L@L and D@L by recrystallization with THF (solid

borderline) and CHCl₃: HEX = 90: 10 (dash borderline) in air at 298 K ($\lambda_{\text{ex}} = 247 \text{ nm}$). (f) The phosphorescence intensity of different doped samples for different solvent systems. (The guest-to-host ratio is 0.1% for all samples in the solid state)

In response to the reviewer's valid critique, we have elaborated on these methodological details within the manuscript to clarify the relationship between our preparation methods, the resulting microstructure, and their impact on the RTP chiral recognition performance. We have redrawn the Figure 4a-4f, revised the corresponding caption to exhibit our new findings and added the following texts in "Exploration of morphology and microstructure" Paragraph: "As can be seen from the scanning electron microscope (SEM) images (Figure 4a-4d) and the power X-ray diffraction (PXRD) patterns (Figure S28), doping with guest molecules of different chirality have no observable effect on the microscopic morphologies for solid-state samples obtained from either tetrahydrofuran (THF, Figure 4a-4b) or a mixture of chloroform/n-hexane (CHCl₃/HEX, Figure 4c-4d). However, the influence of solvent is tremendous: evaporation from the THF solution produces crystals of much smaller sizes with lower aspect ratios, which in contrast to large belt-like crystals acquired from CHCl₃/HEX. The spectroscopic differences were also measured (Figure 4e-4f), where much stronger RTP intensity and a higher ep value are achieved for samples obtained from THF."

These additional details will provide a clearer understanding of the scientific foundation underlying our proposed mechanism and its practical implications.

Comment 3. As shown in Figure 3, in addition to the RTP emission, the intensity ratio between the steady-state PL and delayed emission of fluorescence is also affected by the chirality difference between the host and guest, contrary to the authors' proposal of triplet-triplet energy transfer, as shown in Figure 1c. Explain.

Author Reply 3: We thank the reviewer for the astute observation regarding the intensity ratio differences between steady-state photoluminescence (PL) and delayed emission of fluorescence as depicted in Figure 3. This indeed warrants a detailed explanation, particularly in the context of our proposed triplet-triplet CDET mechanism highlighted in Figure 1c.

The observation that the intensity ratio between steady-state PL and delayed emission is influenced by the chirality of the host-guest interaction does not necessarily contradict the proposed triplet-triplet energy transfer mechanism. First of all, Dexter energy transfer, which could involve chirality-dependent electron hopping, is not exclusive to the triplet-triplet energy transfer; singlet exciton is also subject to chirality selection with the Dexter-type energy transfer. In addition, the Förster-type energy transfer, which is not expected to be chirality-dependent, can also occur between the donor and acceptor singlet excited states. A combination of the two pathways may still co-exist to give lower ep values for fluorescence vs. RTP since no Förster-type is possible for the triplet-triplet energy transfer process. We have indicated the possibility in Reply 4 to Reviewer 2.

Comment 4. The full name of the CDET should be presented when the term is introduced in the article for the first time.

Author Reply 4: We thank the reviewer for the careful review, we have fixed the error and defined the term “CDET” in the main text with color highlighting.

Comment 5. In Figure 2d, the excitation wavelength, monitoring wavelength, and captured delay time should be described in the caption.

Author Reply 5: We have fixed the error and added all necessary information in the caption with color highlighting.

Comment 6. In “Regulation of enantioselective differentiation and mechanism verification.” Paragraph, “.....whereas D-B@L has weak cyan emission with blue afterglow from the host phosphorescence.....” D-B@L should be D-Br@L.

Author Reply 6: We thank the reviewer and are sorry for this mistake. We have made the correction.

Comment 7. I cannot find a corresponding chemical structure to the sample code “L-F-H” either in the manuscript or SI.

Author Reply 7: We are sorry for the mistake, the term “L-F-H” should be “F-Ph-L”, we have corrected this error.

Comment 8. According to Figure S2, the retention time of the D-form and L-form for F-Na appears at approximately 10 min and 15 min, respectively. This is completely different from what Figure S4 displays in the HPLC spectra. However, the authors assigned both of them as the same compound (F-Na). Recheck it!

Author Reply 8: We are sorry for the mistake, Figure S4 should be the HPLC spectra for the retention time of the D-form and L-form for **F-Na-Br**. We have corrected this error.

Comment 9. According to this article, the authors only provide absorption spectra in acetonitrile for all title compounds and lack solid-state absorption data. Since either RTP or CPL emission is observed in the solid state, it is necessary to provide solid-state absorption spectra.

Author Reply 9: We thank the reviewer for their comment regarding the necessity of providing solid-state absorption data, considering that both RTP and CPL emissions were observed in the solid state in our work.

Indeed, we have provided absorption spectra for all title compounds in acetonitrile, which are illustrated in Figure S5 of the supplementary materials. This was to initially characterize the absorption properties of the compounds in a common solvent, allowing for a consistent comparison across different samples.

The solid-state absorption spectra allow us to directly correlate the photophysical behavior of our compounds in the state in which they are utilized for chiral recognition. This information is vital for the interpretation of RTP and CPL phenomena, as the optical properties in the solid state can differ significantly from those in solution due to factors such as intermolecular interactions, crystal packing, and the presence of defects. Recognizing the importance of correlating these data with solid-state properties, we have also conducted solid-state absorption measurements. These results are presented in Figure R4 as well as Figure S6, complementing our liquid-state measurements.

In response to Reviewer 1 (Author Reply 2), we have discussed the implications of these solid-state spectra and have provided a detailed analysis of how the host and guest molecules' absorption characteristics contribute to the observed RTP behavior. These discussions have been integrated into the revised manuscript to provide a comprehensive understanding of the materials' behavior in their functional state.

We hope that the inclusion of this additional data and discussion addresses the reviewer's concerns and substantiates the scientific underpinnings of our findings.

Comment 10. It seems like the authors prepared this manuscript in a rush, where the organization, statements and labels are not well prepared, which has to be trimmed completely before submission elsewhere.

Author Reply 10: We sincerely apologize if the presentation of our manuscript gave the impression of being hastily prepared. The interest generated by our previous publication and the subsequent attention from major news outlets, as well as inquiries from experts in the field, have indeed prompted us to publish our findings promptly to share our results with the scientific community. However, we assure that we have not compromised on scientific rigor.

We recognize the importance of a well-organized manuscript, clear statements, and accurate labels to effectively communicate our research. We are committed to thoroughly revising the manuscript to improve its clarity, coherence, and presentation. We will meticulously review the organization of the content, refine our statements for precision, and ensure that all labels are correctly and clearly presented in the revised manuscript.

REVIEWER COMMENTS

Reviewer #1 (Remarks to the Author):

The resubmitted manuscript has well addressed the problems and it can be accepted now.

Reviewer #2 (Remarks to the Author):

I hold my evaluation that the protocol is novel and interesting, and the characterization is extremely adequate. Also, all my concerns have been carefully replied. Therefore, the manuscript can be accepted as it is in Nat. Commun.

Reviewer #3 (Remarks to the Author):

I have read carefully the replies and revised manuscript. It seems like that authors may address, in part, my comments and make the corresponding changes. However, the main concern regarding the origin of RTP is still unconvincing. It still puzzles me why different guest materials exhibited similar RTP emission wavelength when doped in the same host system (Figure 5). In fact, reviewer2 also raised a similar query, The authors originally proposed a host-guest T-T energy transfer and there is no more further explanation elaborated in the revised text and reply letter. Therefore, I must conclude that the core issue is still not clearly resolved.

Also, in this revised manuscript, the authors provided absorption spectra of the titled compounds depicted in Figure S6. Accordingly, it seems clearly that both host F-Ph-L and guest F-Na-L have absorption at 290 nm and 247 nm. Then, I am curious how the authors are able to excite F-Ph-L only instead of F-Na-L upon e.g., 247 nm excitation? Moreover, as the authors proposed that the RTP emission originated from triplet-triplet energy transfer from host to guest, why the emission spectra of host-guest system are much different at different excitation wavelengths (for 290 nm and 247nm, see Figure 2b and Figure 2c). For example, upon excitation at the long wavelength (290 nm), dual emission with ~350 nm (normal emission) and ~520 nm (RTP) was observed, while only RTP of ~ 520 nm appears upon excitation at 247 nm? Also, RTP of host (F-Ph-L)-guest (F-Na-L) system is different from the emission spectra of pure F-Na-L in the solid state at 77K after a delayed time of 5 ms

(see the comparison between Figure 2 and Figure S8f). These are still all puzzling the reviewer, which must be resolved before the manuscript can be considered for possible publication.

REVIEWER COMMENTS

Reviewer #1 (Remarks to the Author):

The resubmitted manuscript has well addressed the problems and it can be accepted now.

Reviewer #2 (Remarks to the Author):

I hold my evaluation that the protocol is novel and interesting, and the characterization is extremely adequate. Also, all my concerns have been carefully replied. Therefore, the manuscript can be accepted as it is in Nat. Commun.

Reviewer #3 (Remarks to the Author):

I have read carefully the replies and revised manuscript. It seems like that authors may address, in part, my comments and make the corresponding changes. However, the main concern regarding the origin of RTP is still unconvincing. It still puzzles me why different guest materials exhibited similar RTP emission wavelength when doped in the same host system (Figure 5). In fact, reviewer2 also raised a similar query, The authors originally proposed a host-guest T-T energy transfer and there is no more further explanation elaborated in the revised text and reply letter. Therefore, I must conclude that the core issue is still not clearly resolved.

Also, in this revised manuscript, the authors provided absorption spectra of the titled compounds depicted in Figure S6. Accordingly, it seems clearly that both host F-Ph-L and guest F-Na-L have absorption at 290 nm and 247 nm. Then, I am curious how the authors are able to excite F-Ph-L only instead of F-Na-L upon e.g., 247 nm excitation? Moreover, as the authors proposed that the RTP emission originated from triplet-triplet energy transfer from host to guest, why the emission spectra of host-guest system are much different at different excitation wavelengths (for 290 nm and 247nm, see Figure 2b and Figure 2c). For example, upon excitation at the long wavelength (290 nm), dual emission with ~350 nm (normal emission) and ~520 nm (RTP) was observed, while only RTP of ~ 520 nm appears upon excitation at 247 nm? Also, RTP of host (F-Ph-L)-guest (F-Na-L) system is different from the emission spectra of pure F-Na-L in the solid state at 77K after a delayed time of 5 ms (see the comparison between Figure 2 and Figure S8f). These are still all puzzling the reviewer, which must be resolved before the manuscript can be considered for possible publication.

Responses to the comments of Reviewer #3:

I have read carefully the replies and revised manuscript. It seems like that authors may address, in part, my comments and make the corresponding changes. However, the main concern regarding the origin of RTP is still unconvincing. It still puzzles me why different guest materials exhibited similar RTP emission wavelength when doped in the same host system (Figure 5).

Author Reply: We thank the reviewer for raising this question and the simple answer is that there is only one single guest phosphor used for all samples, which is precisely why the same RTP emission could be observed.

As we have shown in Figure 1a, the guest phosphor has two modular units. Specifically, the various amino acids shown in Fig. 5 are NOT directly used as guest molecules. Instead, these amino acids serve as chiral functional groups for the guest molecules, whose emitting moiety is the naphthoyl group, i.e., *the reaction products of various amino acids with 2-naphthoyl chloride constitute the entire guest molecules*. However, amino acids do not exhibit green RTP as they lack intrinsic phosphor; according to literature, tryptophan has the longest emission wavelength for phosphorescence but it exhibits blue phosphorescence at 77 K (Naqvi et al., *J. Photochem. Photobiol. B: Biol.* **2016**, *157*, 120-128).

Despite variations in amino acid types, **the sole RTP emitter within the guest molecules remains the naphthoyl group**. As a consequence, when these different guest molecules are doped into the same host matrix, they are expected to exhibit similar if not the same RTP emission wavelengths. The CDET effect leads to significant intensity differences in the emission spectra of enantiomers, while the emission peak positions remain consistent. This variation across amino acid types precisely confirms the universality of the CDET effect. To address potential misunderstandings, we have revised the manuscript: **All 19 chiral natural amino acids undergo a one-step reaction with 2-naphthoyl chloride to form guest molecules (analytes), and the ep_{RTP} values for a total of 19 chiral amino acid pairs were listed, where 15 of them could be reliably discriminated with an ep value ≥ 3.0 .**

In fact, reviewer2 also raised a similar query, The authors originally proposed a host-guest T-T energy transfer and there is no more further explanation elaborated in the revised text and reply letter. Therefore, I must conclude that the core issue is still not clearly resolved.

Author Reply: Regarding the triplet-triplet energy transfer between host molecules and guest molecules, we have already presented two crucial pieces of evidence in the previous Reply:

(A) Effective quenching of the host photoluminescence. When comparing the phosphorescence emission spectra of the pure host **F-Ph-L** solid and doped samples at 77 K, we observed that when guest molecules share the same chirality as the host molecule, the phosphorescence intensity of the host molecule ($\lambda_{em} = 420$ nm)

significantly decreases, while simultaneously, the phosphorescence intensity of guest molecules ($\lambda_{em} = 526$ nm) significantly increases (Figure R1a-R1b). Furthermore, we measured the phosphorescence lifetime of the host molecule ($\lambda_{em} = 420$ nm) and found that the phosphorescence lifetime significantly decreased when the chirality of the guest and host molecules matched (Figure R1c-R1d). This finding demonstrated that energy transfer between host and guest molecules is more efficient in doped samples with the same chirality, resulting in more effective quenching of the host photoluminescence compared to those with opposite chirality.

Fig. R1 a), b) Delayed emission (DE, $\Delta t = 5$ ms) spectra of two guests in **F-Ph-L** solid (w/w = 0.1% or 1%) in air at 77 K ($\lambda_{ex} = 247$ nm). c), d) Time-resolved RTP emission for two guests in **F-Ph-L** solid (w/w = 0.1% or 1%) at 77 K

(B) Correlation between the ep_{RTP} values and the solid-state absorbance of the host molecules. We also provided new spectra in the previous Reply. When comparing the ep_{RTP} values with respect to the solid-state UV absorption spectra of the host and guest molecules, we found that the ep of fluorescence (ep_{FL}) is independent of absorption wavelengths, while the ep of phosphorescence (ep_{RTP}) is positively correlated with the solid-state absorbance of the host material (Figure R2). This indicates that when the excitation wavelength is < 300 nm, the triplet excited-state energy is initially generated by the host and then transferred to the guest, resulting in a larger ep_{RTP} value for phosphorescence. However, when such an energy transfer process from the host to the guest is no longer possible (*i.e.*, wavelength > 300 nm), the phosphorescence

intensities of **0.1% L@L** and **0.1% D@L** are almost indistinguishable, leading to the ep_{RTP} value close to 1. The evidence is almost irrefutable that the triplet-triplet energy transfer, which is absent in fluorescence, is mainly responsible for the observed phenomenon, verifying the mechanism of the CPE theory.

Fig. R2 a) Solid absorption spectra of the host and guest molecules. b) The wavelength-dependent ep values of the doped samples. (w/w = 0.1%)

It is quite possible that the Reply is lengthy and can become challenging to read.

Also, in this revised manuscript, the authors provided absorption spectra of the titled compounds depicted in Figure S6. Accordingly, it seems clearly that both host F-Ph-L and guest F-Na-L have absorption at 290 nm and 247 nm. Then, I am curious how the authors are able to excite F-Ph-L only instead of F-Na-L upon e.g., 247 nm excitation?

Author Reply: Indeed, both the host molecules and guest molecules exhibit absorption at 247 nm. However, considering that the molar extinction of the host matrix is greater than that of the guest molecule at 247 nm, and given that the host molecules make up more than 99% of the samples, we can safely conclude that the photon energy at 247 nm is mainly absorbed by the host matrix and then transferred to the guest molecules. Notably, the host matrix has a strong absorption at 247 nm, whereas its absorption is significantly weaker at 298 nm (not 290 nm, Figure 2b), leading to the rapid decrease in the ep_{RTP} value from ~ 10 ($\lambda_{ex} = 247$ nm) to ~ 1 ($\lambda_{ex} = 298$ nm), (Figure R2). The variation of the ep value closely aligns with the absorption spectra of the host matrixes. This phenomenon provides compelling evidence that, during short wavelength excitation ($\lambda_{ex} < 300$ nm), energy is initially absorbed by the host matrix before being transferred to the guest molecules.

We would like to remind the reviewer that we also doped the guest molecules into the polymer matrix. Under these conditions, the guest molecules were directly excited and

emitted phosphorescence. Interestingly, the results demonstrated that there is no significant difference in phosphorescence intensity between the enantiomers of the guest molecules (Figure R3). This provides further evidence that the CPE phenomenon can only be achieved through the CDET effect.

Fig. R3 a), b) Delayed emission ($\Delta t = 5$ ms) spectra of the polymethyl methacrylate (PMMA) and two chiral guests (w/w = 1% or 0.1%) in PMMA in vacuo at 298 K ($\lambda_{\text{ex}} = 247$ nm). c), d) Delayed emission ($\Delta t = 5$ ms) spectra of the polystyrene (PS) and two chiral guests (w/w = 1% or 0.1%) in PS in vacuo at 298 K ($\lambda_{\text{ex}} = 247$ nm).

Moreover, as the authors proposed that the RTP emission originated from triplet-triplet energy transfer from host to guest, why the emission spectra of host-guest system are much different at different excitation wavelengths (for 290 nm and 247 nm, see Figure 2b and Figure 2c). For example, upon excitation at the long wavelength (290 nm), dual emission with ~ 350 nm (normal emission) and ~ 520 nm (RTP) was observed, while only RTP of ~ 520 nm appears upon excitation at 247 nm?

Author Reply: It is important to clarify that the difference between Figure 2b and Figure 2c is not due to the use of different excitation wavelengths (247 nm and 290 nm), but rather because Figure 2b represents steady-state emission spectra, while Figure 2c corresponds to delayed emission spectra. In fact, fluorescence emission at 357 nm can also be observed upon excitation at 247 nm (Figure R4). However, using an excitation wavelength of 247 nm would result in a loss of spectral information in the range of 480-560 nm. This interference is due to a significant second-harmonic generation (SHG) peak from the

excitation light itself at approximately 494 nm. Besides, **the ep_{FL} is independent of absorption wavelengths**. Therefore, we opted for an excitation wavelength of 298 nm for the steady-state photoluminescence spectra to ensure spectral integrity across both the fluorescence and phosphorescence ranges.

Fig. R4 Steady-state photoluminescence spectra of the doped samples at 298 K ($\lambda_{ex} = 247$ nm).

Also, RTP of host (F-Ph-L)-guest (F-Na-L) system is different from the emission spectra of pure F-Na-L in the solid state at 77K after a delayed time of 5 ms (see the comparison between Figure 2 and Figure S8f). These are still all puzzling the reviewer, which must be resolved before the manuscript can be considered for possible publication.

Author Reply: Due to the structural similarity between the guest molecules and host molecules, when guest molecules are doped into the host matrix, they exhibit properties akin to single-molecule emission. This behavior resembles the luminescent characteristics of dilute solutions of the guest molecules. Consequently, the RTP emission spectra of the doped samples (Figure 2c) are similar to the phosphorescence emission spectra of dilute solutions of the guest molecules (Figure S8e) at 77 K. However, Figure S8f represents the phosphorescence emission spectra of guest molecules in an aggregated state at 77 K. Therefore, the emission peak in Figure S8f is redshifted compared to that in Figure 2c. (Kasha M. Energy transfer mechanisms and the molecular exciton model for molecular aggregates[J]. *Radiation research*, 1963, 20(1): 55-70.)

In summary, we believe that the evidence for triplet-triplet energy transfer is extremely solid and can be used to account for the theoretical basis for the presented method.

REVIEWERS' COMMENTS

Reviewer #3 (Remarks to the Author):

I have carefully examined the authors' replies and corresponding changes again. I noticed that the reply of my first comment is that the emitter responsible for the emission is ascribed to the naphthoyl group, written below.

"Despite variations in amino acid types, the sole RTP emitter within the guest molecules remains the naphthoyl group."

However, the amino acid described in Figure 5 has no naphthoyl group.

To the reviewer's viewpoint, the mechanism of the CDET system remains unclear at current stage. However, the authors have conducted comprehensive work and the experimental consequence did demonstrate chiral selective RTP. Therefore, the manuscript can be accepted for publication in Nat. Commun. after minor revision elaborated above.

REVIEWER COMMENTS

Reviewer #3 (Remarks to the Author):

I have carefully examined the authors' replies and corresponding changes again. I noticed that the reply of my first comment is that the emitter responsible for the emission is ascribed to the naphthoyl group, written below.

"Despite variations in amino acid types, the sole RTP emitter within the guest molecules remains the naphthoyl group."

However, the amino acid described in Figure 5 has no naphthoyl group.

To the reviewer's viewpoint, the mechanism of the CDET system remains unclear at current stage. However, the authors have conducted comprehensive work and the experimental consequence did demonstrate chiral selective RTP. Therefore, the manuscript can be accepted for publication in Nat. Commun. after minor revision elaborated above.

Responses to the comments of Reviewer #3:

Author Reply: We thank the reviewer for raising this question. As indicated in the revised manuscript, "All 19 chiral natural amino acids undergo a one-step reaction with 2-naphthoyl chloride, resulting in the formation of guest molecules (analytes)." The *in-situ* chemical attachment of the naphthoyl moiety to the amino acids to generate (2-naphthoyl)tyrosine was confirmed by the ^1H NMR spectrum. (Figure R1-R2)

Fig. R1 ^1H NMR spectrum of the crude product of L-Tyrosine and 2-naphthoyl chloride in d-DMSO.

Fig. R2 ¹H NMR spectrum of the crude product of D-Tyrosine and 2-naphthoyl chloride in d-DMSO.

According to the reviewer's suggestion, we have modified Figure 5 by incorporating the chemical structures of the guest and host molecules. (Figure R3) We hope this addition will avoid any potential misunderstandings.

Fig. R3 A universal sensing material for chiral standard amino acids. a) Chemical structure of host and guest in chiral recognition of amino acids. b) $E_{p, RTP}$ values for chiral amino acids grouped into different structural categories, where it is found that a total of 15 pair of chiral amino acids could be reliably distinguished ($ep \geq 3.0$) while the other four failed the test protocol due to poor reactivity between the acyl chloride and the chiral amino acid based on NMR evidence. c) chemical structures with one-letter abbreviations of the tested chiral amino acids, and photographs showing the visual RTP “afterglow” difference using the same sensing medium material under the protocol.